# Microbiota-Derived Extracellular Vesicle as Emerging Actors in Host Interactions

**DOI:** 10.3390/ijms25168722

**Published:** 2024-08-09

**Authors:** Paola Margutti, Antonella D’Ambrosio, Silvia Zamboni

**Affiliations:** Department of Neurosciences, Istituto Superiore di Sanità, 00161 Rome, Italy; antonella.dambrosio@iss.it (A.D.); silvia.zamboni@iss.it (S.Z.)

**Keywords:** bacterial extracellular vesicles, microbiota, dysbiosis, neuroninflammation, immune-cell-response, gut-brain axis

## Abstract

The human microbiota is an intricate micro-ecosystem comprising a diverse range of dynamic microbial populations mainly consisting of bacteria, whose interactions with hosts strongly affect several physiological and pathological processes. The gut microbiota is being increasingly recognized as a critical player in maintaining homeostasis, contributing to the main functions of the intestine and distal organs such as the brain. However, gut dysbiosis, characterized by composition and function alterations of microbiota with intestinal barrier dysfunction has been linked to the development and progression of several pathologies, including intestinal inflammatory diseases, systemic autoimmune diseases, such as rheumatic arthritis, and neurodegenerative diseases, such as Alzheimer’s disease. Moreover, oral microbiota research has gained significant interest in recent years due to its potential impact on overall health. Emerging evidence on the role of microbiota–host interactions in health and disease has triggered a marked interest on the functional role of bacterial extracellular vesicles (BEVs) as mediators of inter-kingdom communication. Accumulating evidence reveals that BEVs mediate host interactions by transporting and delivering into host cells effector molecules that modulate host signaling pathways and cell processes, influencing health and disease. This review discusses the critical role of BEVs from the gut, lung, skin and oral cavity in the epithelium, immune system, and CNS interactions.

## 1. Introduction

It is well established that the human body is a symbiotic ecosystem that contains a diverse range of commensal microorganisms, including bacteria, virus, and fungi, collectively termed microbiota, whose interaction with a human host affects several physiological and pathological processes [1]. Almost every niche of the human body, especially the skin and mucosal surfaces, particularly oral cavity, respiratory, gastrointestinal, and urogenital tracts, is colonized by a large number of microbes [2] (Figure 1). The microbial colonization is especially relevant in the intestinal tract, where over 99% the microbial population mainly consists of bacteria, belonging to six taxonomic groups or phyla, such as *Actinobacteria*, *Bacteroidetes*, *Firmicutes*, *Proteobacteria*, *Cyanobacteria*, and *Fusobacteria* [3]. The role of microbiota in health and disease is being highlighted by numerous studies revealing gut microbiota as a pivotal player in maintaining host health. The relationship of these complex microbial communities with a human host can be either symbiotic (mutually beneficial), commensal (neutral co-existence), or potentially pathogenic [4]. The majority of healthy gut bacteria are non-pathogenic and co-habit with the enterocytes in a symbiotic relationship, supporting host nutrient, xenobiotic, and drug metabolism, preventing pathogenic microorganism colonization and maintaining intestinal barrier function. Several approaches, mainly germ-free models, where animals are reared in a sterile environment so as not to be exposed to any microorganisms, and the manipulation of microbiota, either with antibiotic treatment or microbiota reconstitution, provide key evidence for the role of the gut microbiota in immune homeostasis and disease [5,6,7,8]. Gut microbiota is also known to influence the central nervous system (CNS), contributing to its proper functions [9,10,11], and the blood–brain barrier (BBB) integrity [12,13,14,15,16]. Accumulating evidence has shown that prolonged gut microbial dysbiosis, characterized by alterations in the composition and/or functions of the microbiota, including an increase in pro-inflammatory species and a decrease in anti-inflammatory species [17], has been critically implicated in a wide range of human diseases including inflammatory bowel disease (IBD), cancer, mental health, cardiovascular and immune disorders, and neurodegenerative diseases [18,19,20,21]. Gut dysbiosis, resulting from exposure to lifestyle-related or environmental factors, such as toxins, drugs, antibiotics, and pathogen infection, has a critical impact on increasing intestinal permeability with bacterial product and antigen translocation into the blood and lymphatic circulation, leading to systemic chronic inflammation [17]. The oral cavity has been considered to possess the second most complex microbiota in the human body [22]. Recently, the interest in the oral cavity microbiota has increased, especially considering its proximity to the brain [23]. Research on the oral microbiota has revealed that diet, stress, tobacco consumption, and other lifestyle aspects can dynamically influence oral microbiota function and composition [24,25,26]. It has been suggested that a link exists between the development and/or progression of several systemic pathologies such as cardiovascular diseases, neurodegenerative diseases (such Alzheimer’s disease), autoimmune diseases (such as rheumatoid arthritis), and the occurrence of periodontal diseases caused by an imbalance in the oral microbiota or oral dysbiosis, [27,28]. Oral dysbiosis is caused by a reduction in microbial diversity, and specifically by an increase of pathogenic bacteria and a decrease in commensal health-associated bacteria. Furthermore, recent studies have demonstrated that the oral-to-gut and gut-to-oral microbial transmission can regulate pathogenesis, indicating the presence of the oral–gut microbiota axis. Evidence suggests that over half of bacterial species in the gastro-intestinal system undergo oral–gut translocation, even without pathology [29], though individuals who suffer from cirrhosis, colorectal cancer, and rheumatoid arthritis show more pronounced examples of oral-to-gut bacterial translocation [30,31,32]. Among the many known oral bacteria that can be found in the gut of patients with gastrointestinal diseases are members of the genera *Staphylococcus*, *Porphyromonas*, *Veillonella*, *Fusobacterium*, *Actinomyces*, and *Parvimonas* [33]. Indeed, it has been hypothesized that the oral bacteria could transfer to the gut through two different ways: hematogenous and enteric routes. In the hematogenous route, bacteria enter the systemic circulation through oral lesions while in the enteral route, bacteria migrate from the oral cavity via the stomach into the intestines [33].

With the application of next-generation sequencing (NGS) along with omics technologies, recent studies have also focused on the lung and skin microbiota composition related to various pulmonary and cutaneous diseases, respectively [34,35]. In particular, the healthy upper respiratory tract is largely colonized with bacteria whose species composition differs between the nasal cavity and nasopharynx. The nasal cavity shows the presence of *Moraxella*, *Staphylococcus*, *Corynebacterium*, *Haemophilus*, and *Streptococcus* species, while the oropharynx exhibits high abundance of *Prevotella*, *Veillonella*, *Streptococcus*, *Leptotrichia*, *Rothia*, *Neisseria*, and *Haemophilus* species [36]. By contrast, the healthy lower respiratory tract, comprising the trachea and lungs, exhibits a very low concentration of resident bacterial species belonging to the genera *Prevotella*, *Veillonella*, and *Streptococcus*, allowing for the performance of its most crucial function, such as the exchange of oxygen and carbon dioxide [37,38,39]. Particularly, the nasal microbiota, which is distinct from the oral microbiota, contributes little to the composition of the healthy lung microbiota, which derives mainly by the microaspiration of the oropharyngeal or gastro-oesophageal contents. Furthermore, in the lower respiratory tract, the low bacterial concentration derives by a rapid microbial clearance that occurs via a different physiological mechanism, including cough, mucociliary transport, and innate immunity, mainly performed by alveolar macrophages, resulting in dynamic microbiota changes [35,40]. It has been reported that often, the alteration of the composition and function of both intestinal and lung microbiota is associated with chronic respiratory diseases, including asthma and chronic obstructive pulmonary disease (COPD). Indeed, alterations in the gut microbiota are linked with changes in lung immunity and on the other side, the lung microbiota also influences the microbiota in the gut, reflecting the close connection between the intestinal and pulmonary axes [41]. The skin, which plays the body’s highest function as a physical barrier, exhibits biogeographically distinct regions showing different microbial community structures influenced by microenvironments, such as skin pH, temperature, sebum content, and moisture levels. A large proportion of skin microbiota comprises resident bacteria that are generally stable, although there is a smaller percentage of transient microbes that can opportunistically colonize niches when the skin is compromised [42], as occurs with skin ageing, pathology, and injury. This condition can cause dysbiosis, leading to some skin diseases, such as atopic dermatitis. Recently, the omics technologies on microbiota, mainly of the gut as well as the oral cavity, lung, and skin, have increased the knowledge of the interrelationship between a host and bacteria, which has implications for health and disease. Numerous mechanisms exist for bacterial cell communication with other bacteria and host cells, including direct cellular adhesion, the release of cell wall components, and the secretion of metabolically functional products (Figure 2).

Emerging research recognizes that bacteria release extracellular vesicles (BEVs) that are considered as bioactive cargo delivery modes for mediating the interkingdom crosstalk, allowing for the long-distance delivery of bacterial effectors. BEVs, released by pathogenic and commensal Gram-positive and Gram-negative bacteria, are spherical membrane-enveloped particles ranging in size from 20 to 400 nm that disseminate part of the biological content of the parent bacterium into the extracellular milieu. In particular, BEVs carry a range of cargo of bioactive molecules, including periplasmic and cytoplasmic proteins, toxins, metabolites, and nucleic acids, affecting a variety of biological processes, including virulence, horizontal gene transfer, biofilm formation, phage infection, transport of cell metabolites, and bacterial–bacterial or bacterial–host interactions [43,44]. Mounting evidence shows that BEVs can follow different formation routes, which can lead to distinct BEV subtypes with different molecular cargos and, thus, potentially different biological functions [43,45]. According to numerous studies, BEVs—especially those originating from Gram-negative bacteria—play a role in several inflammatory diseases, such as sepsis, gastrointestinal inflammation, periodontal disease, and lung and skin inflammation, by triggering pattern recognition receptors (PRRs), activating inflammasomes and inducing mitochondrial dysfunction. BEVs also affect inflammation in distant organs or tissues via a long-distance cargo transport into the brain. The knowledge of the structure, molecular cargo, and function of BEVs has been obtained primarily from bacteria cultured in laboratory conditions. In contrast, there are very few studies on BEVs in human body fluids probably due to the difficulties in separating BEVs from host-derived extracellular vesicles, such as exosomes and ectosomes.

## 2. Origin and Composition of BEVs

BEVs released by both Gram-negative and Gram-positive bacteria can be generated through membrane blebbing or explosive cell mechanism. Gram-negative bacteria follow two main pathways for vesicle formation. The first formation route involves the blebbing of the outer membrane of the bacterial envelope, generating outer-membrane vesicles (OMVs); the second pathway, occurring when the inner membrane protrudes due to weaknesses in the peptidoglycan layer, involves prophage endolysin-mediated explosive cell lysis forming outer-inner membrane vesicles (OIMVs) and explosive outer-membrane vesicles (EOMVs) [46]. Gram-positive bacteria produce cytoplasmic membrane vesicles (CMVs) lacking an outer membrane. CMVs are released via cell membrane budding and cell wall re-modeling caused by endolysin-mediated degradation of the peptidoglycan, resulting in cell lysis, where the vesicles are extruded through gaps in the rigid cell wall [47]. It is possible that there are additional, non-lethal related mechanisms for CMV production that are not yet explored [45]. The membrane composition of BEVs reflects the envelope architecture of parental bacteria from which BEVs are derived [48]. The cell wall of Gram-negative bacteria consists of a thin layer of peptidoglycan (a polymer-like mesh made of sugars and amino acids) in the periplasmic space between two membrane bilayers: the inner (or cytoplasmic) and outer membranes. The outer membrane contains lipopolysaccharides (LPS also known as endotoxin) on its outer leaflet and various membrane-bound proteins. In contrast, Gram-positive bacteria completely lack an outer membrane but have a much thicker peptidoglycan cell wall, which is linked to the underlying cytoplasmic membrane via lipoteichoic acids (LTA) [49] (Figure 3). The growth and environmental conditions of the parental bacteria as well as the mechanism of BEV biogenesis modulate the composition of their cargo affecting bacteria-bacteria and bacteria–host interaction.

The protein content of BEVs includes structural proteins, porins, ion channels, transporters, enzymes, and proteins related to stress response. The most enriched terms are related to energy generation, such as aerobic respiration and the tricarboxylic acid cycle, and the most abundant enzymes belong to the oxidoreductase family [50]. The size, protein composition, and cargo selection for BEVs can depend on the stage of bacteria originating from BEVs. For instance, research has investigated *Helicobacter pylori*-derived OMVs (Hp-OMVs) isolated and purified from bacterial cultures in early, late, and stationary growth phases [51]. During the growth phase, *Helicobacter pylori* (*H. pylori*), a major pathogen causing chronic inflammation of the gastric epithelium cells, releases OMVs that are crucial to the pathogen–host interactions [52]. Hp-OMVs have been studied both in vivo and in vitro in gastric biopsies [53,54,55], in gastric juice samples of infected individuals [56], and in culture media of *H. pylori*, especially during the late stationary growth phase and during biofilm formation [57,58,59,60]. Specifically, it has been shown that the release of Hp-OMVs is inversely related to cell growth: during the logarithmic phase, OMVs are released in a low quantity; meanwhile, during the stationary phase, OMVs are released at high levels [61]. Furthermore, Hp-OMVs isolated in the early growth phase are enriched in metabolic proteins and virulence factors, such as vacuolating cytotoxin A (VacA), a toxin inducing intracellular vacuolation [62], compared to OMVs from later-stage bacteria [51]. According to epidemiological studies, there is an association between *H. pylori* infection and iron metabolism. Interestingly, iron availability affects the composition of Hp-OMVs [63]. Hp-OMVs obtained from iron-containing media are LPS enriched, whereas Hp-OMVs from bacteria grown under iron-limiting conditions have less and shorter LPS [63]. In addition, a lack of iron inhibits the growth of bacteria but has no effect on the release of Hp-OMVs. It has been observed that OMVs generated under iron-deficient circumstances contain less VacA toxin as demonstrated by the absence of cytoplasmic vacuolization seen in HEp-2 cells following incubation with these Hp-OMVs [64].

OMVs released from *Porphyromonas gingivalis* (*P. gingivalis*) retain the full set of outer membrane constituents, including LPS, muramic acid, a capsule, and fimbriae, reportedly to mediate bacterial adherence to and entry into periodontal cells and proteases termed gingipains, which contribute to the destruction of periodontal tissues [65]. Recently, key differences in the proteome of *Pseudomonas aeruginosa*-derived OMVs generated by different mechanisms of biogenesis, such as OMVs produced naturally, by budding only, or predominately by explosive cell lysis, have been determined [66]. All OMVs, independently of the biogenesis process, contained a subset of proteins that were not found at significant levels in their parent bacteria, as revealed by proteomic analysis results. Particularly, budding OMVs were significantly enriched in the peptidoglycan, proteolysis enzymes, and numerous proteins with functions of siderophore transport and metal binding, suggesting that OMVs produced by budding may contribute to degrading bacterial material in the environment and to acquiring essential metal ions. OMVs produced by explosive cell lysis were significantly enriched in multidrug resistance proteins, suggesting that this OMV type contains a range of cargos that may aid in the survival of bacteria within their environment [66]. Indeed, diverse roles are now being attributed to BEVs, including functions in host pathogenesis and colonization [67], competition with other bacteria [68], antimicrobial resistance [69], quorum sensing [70], horizontal gene transfer, and biofilm formation [71].

In BEVs from different Gram-negative pathogenic bacteria (*Pseudomonas aeruoginosa*, *Porphyromonas gingivalis*, *Salmonella typhimurim*), chromosomal DNA is primarily surface-associated (or extraluminal), with lesser amounts detected in the intraluminal region [72]. Intraluminal BEV DNA sequencing revealed the enrichment of specific bacteria in chromosome regions related to metabolism, stress response, antibiotic resistance, and virulence. The surface-associated versus intraluminal BEV DNA may have different functions. Growing evidence suggests that extraluminal DNA plays a role in biofilm formation; meanwhile, internal BEV DNA is involved in the communication and horizontal gene transfer of antibiotic resistance or virulence genes. The evidence of the BEV-derived DNA detection in the nucleus of human non-phagocytic cells (e.g., epithelial cells) [72] suggests the probability that bacterial DNA could be transferred to the nucleus and integrated into host genome. In fact, the integration of bacterial DNA sequences has been found more frequently in human cancer cells than in normal cells, particularly in tumors connected to the gastrointestinal tract, suggesting a possible function for BEVs containing DNA in the development of cancer [73]. In light of this, it has been reported that Hp-OMVs cause genomic instability in epithelial cells, as assessed using the cytokinesis-block micronuclei assay, a standardized method used for genotoxicity studies, [74], but to date there is no mechanistic study investigating how BEVs can impact oncogenesis and tumor progression. Emerging evidence suggests that BEVs contain differentially packaged noncoding regulatory small RNAs (sRNAs) and transfer RNA (tRNA) fragments, regulating recipient cell gene expression and function. In particular, sRNAs are heterogeneous in size (~20 to 500 nt) and regulate gene expression by base-pairing with the translation initiation region or coding sequence of target host mRNAs [75], suggesting that sRNAs have a function similar to the regulatory role of eukaryotic miRNAs [76]. tRNAs are 70–100 nt long molecules with highly conserved sequences that form secondary cloverleaf and L-shaped three-dimensional structures [77]. Numerous non-canonical roles of tRNAs have been identified, in addition to their well-known function as amino acid carriers to decode mRNA sequences in eukaryotes and procaryotes. Accordingly, tRNA fragments can also mediate gene silencing through an Argonaute-microRNA like mechanism and both positive and negative effects on the global regulation of protein translation [78]. Otherwise, it has been proposed that sRNA fragments could control the expression of target cell genes by base pairing with the target or by sequestering regulatory proteins [79]; however, many aspects are still unknown. Few studies have found particular sRNAs that mediate the host response to BEVs, despite recent studies demonstrating that sRNAs in BEVs affect host cell biology [80,81]. Although recent studies have shown that sRNAs in BEVs affect host cell biology, to date very few studies have identified specific sRNAs that mediate the host response to BEVs [80,81]. Furthermore, a large number of sequences of the transported small noncoding RNAs within BEVs derived from *Escherichia coli* align to regions of the human genome that are involved in the regulation of gene expression linked to epigenetic mechanisms (chromatin remodelling, histone modifications) or cell-specific transcriptional control [82]. Recent studies have also demonstrated that host cells secrete EVs containing miRNAs that regulate the gut microbiota, biofilm formation, and the bacterial response to antibiotics. On the other hand, the secretion of noncoding regulatory sRNAs delivered by BEVs suggests that regulatory RNAs in bacterial or host extracellular vesicles mediate a bidirectional communication between bacteria and host cells [83]. Research on the complete characterization of the molecular composition of the luminal and membrane compartments of BEVs, as well as how these contents vary in response to stressors or environmental cues, may shed light on BEVs function in a variety of physiological and pathological processes.

## 3. BEVs in Systemic Circulation

Under healthy and pathological conditions, BEVs have been found to translocate to the lymphatic or blood circulation, allowing them to interact with various host cells within the body [84]. Initial research has established the presence of circulating gut microbiota-derived BEVs in the plasma of patients with altered intestinal barrier function [85], but mounting evidence indicates that intestinal BEVs pass through mucosal barriers also under healthy conditions [86]. Hence, a study aiming to assess the presence of OMVs in blood samples of patients with well-defined intestinal barrier dysfunction, who had been diagnosed with IBD, HIV, and intestinal mucositis by measuring systemic LPS associated with OMVs, showed increased levels of LPS in the plasma from these patients than healthy subjects. Moreover, patients with altered epithelial barrier presented higher plasma levels of zonulin, a marker of impaired barrier integrity which causes tight junction (TJ) disassembly by phosphorylating ZO proteins [76]. Furthermore, in this study, the paracellular translocation of BEVs was also demonstrated in an in vitro colitis model where TJ integrity was compromised. Recently, it has also been demonstrated that there is a presence of BEVs in systemic circulation of healthy subjects. In particular, total host endogenous extracellular vesicles (EVs), comprising exosomes and ectosomes, purified from the red blood cell concentrates of healthy donors revealed variable amounts of the outer membrane protein A (OmpA) and LPS in a large majority of samples analyzed, providing indirect experimental evidence for the presence of gut microbiota-derived BEVs in human circulating blood in the absence of intestinal barrier disruption [86]. Furthermore, through immunoblotting analysis of total EVs for detecting LPS and LTA expression as markers of Gram-negative and Gram-positive bacteria, respectively, it has been demonstrated that circulating BEVs increased across the human lifespan and were associated with age-related increased intestinal permeability [87]. There are strong indications that systemically circulating BEVs can cause relevant clinical impact in different locations of the human body. Mouse models studies based on BEV-labeling strategies, following intraperitoneal, tail, and oral administration, allow for the exploration of BEV biodistribution and location in systemic organs. A recent study looked into the biodistribution of fluorescently labeled BEVs derived from the major human gut commensal bacteria, *Bacteroides thetaiotaomicron* (Bt), a Gram-negative anaerobe, as a major constituent of the human cecal and colonic microbiota, following oral administration in mice under normal healthy conditions [88]. Using a combination of in vitro culture systems including intestinal epithelial organoids and in vivo imaging, commensal Bt-OMVs have been found to be internalized by intestinal epithelial cells via several endocytic routes.

Considering the size selectivity of each endocytosis route, the use of distinct pathways by Bt-OMV uptake suggests their potential size heterogeneity (20 to >400 nm). This is supported by the recent discovery that the size of Hp-OMVs determined their endocytosis mechanism [52,89]. Moreover, most endocytic routes of Bt OMV uptake culminated in lysosomes located in a peri-nuclear region [90,91] without autophagosome formation and, therefore, autophagy, a lysosome-mediated degradative system essential for cell homeostasis [92]. This evidence was found using small intestinal organoid monolayers generated from mice of both wild-type control and mutant knockout for Atg16l1 protein, a key component of the canonical autophagy pathway, where the peri-nuclear localization of fluorescent DiO-labeled Bt OMVs was comparable in both wild type and Atg16l1-deficient organoid monolayers, excluding any role of autophagy as a cellular process in Bt-OMV intraepithelial trafficking [88]. In addition to cellular uptake, using small intestinal and cecal organoid epithelial monolayers incubated with DiO-OMVs for 24 h, it has been shown that a proportion of Bt-OMVs transmigrated through host intestinal epithelial cells via a paracellular route in vivo to reach the lamina propria for accessing underlying immune cells, the vasculature, and systemic tissues [88]. Collectively, these data suggest that Bt-OMVs transiently modulate the host TJ barrier in order to transmigrate across the intestinal epithelium via the paracellular pathway. Moreover, using an in vivo model where fluorescent DiD-labeled Bt-OMVs were orally administered to mice for 8 h prior to organ excision, BEVs have been found in systemic tissues [88].

In particular, the highest signal was detected in the gut lumen, mainly in the small intestine (51.35%), stomach (11.71%), caecum (19.70%), and colon (10.14%). Lower intensity signals were evident in systemic tissues including the liver (9.99%), lungs (1.04%), and heart (0.63%), suggesting that Bt-OMVs can translocate through the intestinal epithelium to reach various systemic tissues, entering the circulatory or lymphatic systems, with the greatest accumulation in the liver [88], reached probably through the portal vein.

In another study aiming to investigate the journey and spreading of OMVs after their administration in healthy mice, an engineered non-pathogenic *Escherichia coli* (*E. coli*) strain capable to produce homogeneous OMVs containing miRFP713, an easily detectable and quantifiable fluorescent protein derived from *Rhodopseudomonas palustris*, was used for non-invasive and deep-tissue in vivo imaging [93]. miRFP713-OMVs injected into the tail vein of three mice rapidly distributed broadly through the blood circulation into the whole mouse body where they could be detected in various organs up to 24 h post-injection. In particular, after 1 h of miRFP713-OMV caudal intravenous injection, an intense signal of fluorescent was observed in the blood and the bladder, suggesting that a substantial fraction of the OMVs was rapidly directed to the kidneys and then the bladder for elimination with urine. However, the fluorescence levels detected ex vivo in spleen cells after 5 h OMV administration in the three mice showed the fluorescent signals in monocytes and dendritic cells (DCs) by flow cytometry. In particular, the miRFP713 signal was maintained for 24 h after OMV injection in monocytes, whereas in DCs, the fluorescent signal was stronger at 5 h than 24 h post-intravenous injection, suggesting that the cells that had taken up the OMVs had either degraded the miRFP713 protein or had migrated to another site [94]. Furthermore, it was also found that the miRFP713-OMVs underwent biodistribution after their oral gavage administration. The results showed significant signals in the skin, in the stomach, and in the digestive system 5 h after the first gavage and a significant signal only on the skin 24 h after the 3rd gavage, suggesting the fluorescent diffusion to distant organs and their resistance to the gastric environment [94]. The finding that fluorescent OMVs were detected in mice spleen cells, such as monocytes and DCs, after their tail vein injection, was confirmed by another study aiming to explore the OMV interaction with human peripheral blood mononuclear cells (PBMCs) purified from healthy donors [86].

When PBMCs, containing several cell populations including monocytes, T-cells, B-cells, NK-cells, and the γδ T cells that are unconventional T-cells expressing the γδ TCR known to recognize bacterial components [95], were incubated with DiD-labeled OMVs isolated from a culture of *E. coli*, it was observed that [96] monocytes almost exclusively showed significant fluorescent signal [86]. In conclusion, the main route used by phagocytic cells of the immune system, such as neutrophils, macrophages, and DCs to internalize BEVs is phagocytosis, meanwhile in non-phagocytic cells, such as epithelial cells, BEVs are internalized by different endocytic pathways, mainly including clathrin-mediated endocytosis and lipid raft-mediated processes that may depend on caveolin [89]. These findings suggest that BEVs, released by gut bacterial species into the intestinal lumen, cross the epithelial barrier preferentially via a paracellular route under gut dysbiosis, which is associated with epithelial barrier disruption [88]. In healthy conditions, gut microbiota-derived BEVs can be internalized into epithelial cells by different endocytic pathways and might be translocated into the lamina propria by transcytosis or also via a paracellular route whether BEVs might be able to transiently modulate the host TJ barrier. BEVs translocated into lamina propria are able to interact with various resident immune cell populations (DCs, neutrophils, and macrophages) as well as potentially disseminate more widely around the body via the systemic or lymphatic circulation to reach distant tissues and organs, potentially including the brain. The detection of bacterial nucleic acids in body fluids, such as blood and CSF, can be justified in view of the capability of gut BEVs carrying DNA to translocate into systemic circulation [97,98]. In a transgenic mouse model of Alzheimer’s disease, the 16S rRNA metagenomic analysis of BEVs in blood revealed taxonomical diversity reflecting the diversity of the intestinal microbiota, allowing for identifying more than 3000 taxonomic units that correlated with the gut microbiota profile [99]. Likewise, the genomic profile of BEVs isolated from urine samples of patients with autism disorder showed great differences compared to healthy controls, reflecting the changes in gut microbiota [100]. In light of these findings, blood-derived BEVs could represent an alternative to fecal sampling for profiling the gut microbiota and evaluating pathogenic variations in the intestinal microbiota (dysbiosis) in the context of neurodegenerative diseases and, perhaps, various other pathological conditions. Although gut microbiota-derived BEVs have been discovered in blood and urine samples of healthy subjects and the optimized protocol design for their recovering, combining the sequential application of size-exclusion chromatography and density-gradient ultracentrifugation, from human blood has been reported in detail, until now, studies based on BEV–host interaction using BEVs purified from body fluids have not yet been conducted [101]. The use of BEVs purified directly from the blood would be of great help to better understanding their functions and assessing their application as possible diagnostic tools.

## 4. BEV Interaction with Epithelial Cells

Epithelial cells form the interface between the body and the external environment in multiple sites, including airways, the oral cavity, and the gastrointestinal tract, which are in direct contact with microbiota. These cells, considered the first line of defense against inorganic, organic, and pathogen invaders, are important guardians detecting dangers and initiating defense responses. Epithelial cells recognize various microbe-associated molecular patterns (MAMPs) through diverse sets of receptors including Toll-like receptors (TLRs), generally expressed at lower levels than in professional immune cells, and nucleotide-binding oligomerization domain-containing proteins (NODs). TLR signaling induces multiple pathways to activate various inflammation-relevant transcription factors, including the nuclear factor-κ B family of factors (NF-κB), members of the interferon regulatory factor family, and activator protein 1, which lead to the expression of various cytokines, chemokines, interferons, and anti-microbial molecules. MAMPS, which constitute the majority of the signaling molecules, are found within BEVs. In general, LPS is recognized by TLR4, lipoprotein binds to TLR2, and microbial DNA and RNA are recognized by TLR7 and TLR9, respectively. NOD1 and NOD2 [102], as innate immune sensors of bacterial peptidoglycan fragments carried by BEVs, are expressed both by epithelial and immune cells. While NOD2 recognizes muramyl dipeptide, which is ubiquitously present in all bacterial peptidoglycan molecules, NOD1 detects D-glutamyl-meso-diaminopimelic acid, which is mostly present in the peptidoglycan of Gram-negative bacteria [103,104,105]. Activated NOD1 interacts with the specific kinase RIP2, which leads to NF-κB activation and the subsequent upregulation of host genes involved in the inflammatory responses. In particular, there is evidence on the involvement of innate immune sensor NOD receptors in the molecular mechanisms and signaling pathways activated by microbiota BEVs in intestinal epithelial cells that initiate the innate immune response. These cytosolic receptors are crucial for maintaining intestinal homeostasis, playing relevant roles in host defense responses against bacterial pathogens and the regulation of the inflammatory response to microbiota [103,106].

Studies based on mechanisms by which BEVs enter and affect epithelial cells were performed by using Caco-2 and T84 epithelial cell lines as in vitro model systems [107]. In particular, for in vitro cell uptake studies, chemical inhibitors of the three main pathways of endocytosis are used, and then the finding is further confirmed using siRNA to knockdown all three pathways of cellular entry. The main mechanisms by which BEVs can enter non-phagocytic epithelial host cells are membrane fusion, clathrin-mediated endocytosis, caveolin-mediated endocytosis, lipid raft-mediated endocytosis, and macropinocytosis. The internalizing pathway utilized by BEVs to enter epithelial cells depends on their size and composition [108]. It has been found for OMVs released by *H. pylori* that smaller Hp-OMVs, ranging from 20 to 100 nm in size, preferentially enter host cells via caveolin-mediated endocytosis, whereas larger Hp-OMVs, ranging between 90 and 450 nm in size, enter host epithelial cells via macropinocytosis and endocytosis [56]. OMVs released from *Enterohemorrhagic E. coli* (EHEC) enter host cells via clathrin-mediated endocytosis [91], whereas OMVs from *Enterotoxigenic E. coli* (ETEC), *Porphyromonas gingivalis* (*P. gingivalis*), and *Pseudomonas aeruginosa* (*P. aeruginosa*) are internalized through a lipid raft-mediated pathway in a clathrin-independent manner [90,109,110]. In light of all studies focusing on BEV entry into epithelial cells, it has been observed that the uptake of pathogen-derived OMVs by epithelial cells occurs through different pathways depending also on the cargo of the vesicles to be internalized. In particular, the two main endocytic pathways, clathrin-mediated endocytosis and the lipid raft-mediated pathways, produce endosomal compartments with different surfaces that allow for the delivery of their cargos to various subcellular destinations [111]. OMVs released by pathogenic bacteria such as *H. pylori*, *P. aeruginosa*, and *Vibrio cholera* (*V. cholera*) have been suggested to contribute to the pathology of chronic inflammatory diseases, eliciting IL-8 production from gastric [112], bronchial [113], and intestinal epithelial cells [114], respectively. IL-8 released from epithelial cells is a pro-inflammatory cytokine leading to neutrophil influx and inflammation [115]. Increasing evidence shows that Gram-negative pathogens-derived OMVs, once internalized in the host target cells, contribute to virulence by delivering cytotoxic factors, toxins, and mediators that interfere with the immune system [116,117,118].

Moreover, BEVs can carry molecules on their surface, such as toxins, which mediate adhesion to host extracellular proteins or specific cell receptors of epithelial cells determining the primary interaction with target cells, and, consequently, with the derived effects [119]. Indeed, it has been shown that larger Hp-OMVs carrying bacterial adhesion proteins, which are absent in smaller OMVs, are capable of entering epithelial cells efficiently via receptor-mediated endocytosis [120,121]. Furthermore, it has been reported that OMVs less than 100 nm in diameter induce higher levels of NF-κB activity than larger OMVs, suggesting that the smaller OMVs may be more efficient at entering host epithelial cells and initiating pro-inflammatory responses [106]. BEVs can carry molecules on their surface, like toxins, which mediate adhesion to host extracellular proteins or specific cell receptors of epithelial cells, hence dictating the initial interaction with target cells and subsequent consequences [119]. OMVs from pathogenic strains such as *P. aeruginosa* [113], *P. gingivalis* [122], and pathogenic *E. coli* [91,109] have been shown to contain various active virulence factors such as toxins, proteases, and adhesins, in addition to LPS, affecting deeply targeted cells, including epithelial cells. In particular, some toxins, such as EHEC cy-tolysin ClyA, ETEC cytolethal distending toxin V, ETEC heat-labile enterotoxin (LT), *Shigella enterotoxin 1* (ShET1), and *Campylobacter jejuni* (*C. jejuni*) cytolethal distending toxin, seem to use OMVs exclusively as a secretory pathway [91,123] (Table 1).

### 4.1. BEV Interaction with Gastrointestinal Epithelium

The intestinal epithelial barrier is composed of a single layer of polarized epithelial cells, with distinct apical microvilli and crypts, and interconnected to each other by different intercellular types of junctions, consisting of tight junctions (TJ), adherens junctions (AJ), desmosomes, and gap junctions [124]. The intestinal epithelial barrier provides the first line of defense against pathogens, where a mucus layer protects the epithelial surface preventing the intimate contact between epithelium and luminal bacteria [125]. BEVs may significantly impact the intestinal epithelial barrier function via TJ proteins or host inflammatory response regulation. BEVs produced from the gut microbiota may enter the mucus layer directly and be endocytosed by host cells, mediating communication cross-talk between gut bacteria and intestinal epithelial cells. Particularly, Hp-OMVs can be internalized by the gastric epithelium and then promote the destruction of the mucin barrier and the consequent bacterial colonization, inducing inflammation. This effect of Hp-OMVs is dose-dependent, affecting epithelial cell proliferation and inducing the release of IL-6, TNF-α, and IL-8 [56]. Otherwise, more recent studies have demonstrated that OMVs derived from probiotic bacteria, such as *E. coli Nissle 1917* (EcN) [126] and commensal bacteria, such as *Bacteroides fragilis* [127], *B. thetaiotaomicron* [128], and *E. coli* strain ECOR12 [126], modulate the low-level inflammation, crucial for intestinal homeostasis. In particular, *E. coli* C25-derived OMVs elicit a moderate proinflammatory response via the secretion of IL-8 from the colonic carcinoma cell lines HT29-19A and Caco-2 [129]. During infection, EHEC, causing diarrhea and hemolytic uremic syndrome, releases OMVs as-sociated with hemolysin (hly), a pore-forming cytolysin toxin that kills target cells by inserting themselves into the cell membranes, which results in pore formation and, ultimately, cell lysis [91]. Using CaCo2 cells, it has been observed that OMV-associated EHEC-hly, after their internalization into cells through clathrin-dependent endocytosis, enter the endosomal compartments. During endosomal acidification, EHEC-Hly escapes from the lysosomes, targeting mitochondria. The presence of EHEC-Hly in mitochondria leads to a reduction in the mitochondrial transmembrane potential and release of cytochrome c to the cytosol, with subsequent activation of caspase-9 triggering the apoptotic cell death [91]. Hence, OMV-mediated delivery represents a novel mechanism for a bacterial toxin to enter the host cells in order to subvert mitochondrial function, thereby causing cell death. By using Y1 and HT29 mice’s adrenocortical tumor cell lines exhibiting epithelial morphology, researchers have explored the entry into epithelial cells and the pathogenic role of OMV associated with heat-labile enterotoxin (LT) released by ETEC, a prevalent cause of traveler’s diarrhea and infant mortality in third-world countries [109]. LT, comprising an enzymatically active A subunit (ADP-ribosylating) linked to five receptor binding B. subunits, is a major virulence factor which disturbs the electrolyte balance in the host by causing the efflux of water and electrolytes from epithelial cells into the lumen of the intestine [130]. LT has been found to be both inside and on the external part of OMVs [130] and the LT external OMV, binding to ganglioside GM1 receptors present in lipid rafts, which causes the internalization of OMV-LT associated with a cholesterol-dependent pathway of endocytosis [109] that accumulates in a non-acidified compartment of the host cell [98]. Upon LT delivery, the A subunit is then activated, causing elevated levels of cyclic adenosine monophosphate (cAMP), then resulting in secretion of chloride ions and impaired absorption of sodium ions in epithelial cells [109]. OMVs can directly induce apoptosis in intestinal epithelial cells.

In macrophage/Caco-2 co-cultures, *Fusobacterium nucleatum*-derived OMVs promoted epithelial barrier loss and oxidative stress damage, which were associated with epithelial necroptosis or inflammatory cell death, caused by receptor-interacting protein kinase 1 (RIPK1) and receptor-interacting protein kinase 3 (RIPK3) activation. This phenomenon was also confirmed in mouse colitis models [131,132]. Additionally, some OMV cargos from intestinal bacterial pathogens may destroy connections and adhesions be-tween intestinal epithelial cells. In a recent study, OMVs from *E. coli* BL21 were incubated with three different epithelial cell lines, Caco-2, HT29, and NCM460, and, after OMV internalization, they released LPS into the cytosol. Intracellular LPS activated caspase-5, which, in turn, downregulated E-cadherin expression and caused intestinal barrier dysfunction [133]. Indeed, accumulating evidence shows that after their internalization in human intestinal epithelial cells, OMVs release LPS into the cytosol [133]. It has been suggested that early endosomes are the sorting stations for internalized cargos of OMVs that are routed to the cell surface or cytosol, where they exert their biological activity. Early endosomal escape allows for OMV-bound LPS to reach cytosol functionally intact and avoid complete degradation in the lysosomes, facilitated by sorting nexin 10 (SNX10), a member of sorting nexin (SNX) family proteins, acting as a crucial adaptor protein in endosome/lysosome, which activates caspase-5 (in humans). Indeed, this leads to Lyn phosphorylation, subsequently down-regulating E-cadherin expression and impairing the intestinal barrier [133]. Confirming these findings, research has found an accumulated Gram-negative bacteria in the inflamed lesion of intestinal epithelium in the patients with intestinal inflammatory diseases [134], indicating that the elevated LPS from Gram-negative bacteria may engage in the initiation and aggravation of intestinal inflammation via OMVs carrying LPS.

Evidence suggests that some gut pathogenic bacteria can damage the intestinal barrier via BEVs. It has been shown that some enteric pathogens can induce permeability defects in gut epithelial by altering TJ proteins, mediated by their toxins. Reduced TJ integrity greatly increases ion conductance across the paracellular route compared to the transcellular route, resulting in a phenomenon described as leaky gut [135]. This condition basically enables pathogens and bacterial products including BEVS to access in large quantities the whole body by systemic circulation.

In general, changes in gut permeability can be induced via modulation of TJs (down- or up-regulation of the TJ proteins) or their relocation or/and cytokine and hydrogen peroxide-induced decrease in trans-epithelial tissue resistance [136,137]. Enteric bacterial pathogens can target the intercellular tight junctions disrupting them either directly by affecting specific TJ proteins or indirectly by altering the cellular cytoskeleton [138]. TJ barrier disruption of increased paracellular permeability can lead to the activation of the mucosal immune system, resulting in tissue injury and prolonged inflammation.

An altered intestinal permeability, associated with several chronic conditions, has led to an intensified interest in functional studies on the intestinal barrier in different conditions for understanding the pathogenic mechanisms of epithelial barrier disruption in which BEVs are involved. Experimentally, TJ barrier integrity and permeability in intestinal tissues and cells are evaluated by measurement of transepithelial electrical resistance (TER) and the paracellular passage of small molecules, such as mannitol, dextran, and inulin [139]. The infection of *C. jejuni*, a Gram-negative bacterium associated with severe inflammatory enteritis, is a multistep process including colonization of the intestinal mucosa and interactions with and invasion of the human intestinal epithelial cells. Recently, it has been shown that *C. jejuni*-derived OMVs (Cj-OMVs) have a crucial role in bacterial pathogenesis [140]. In particular, it has been demonstrated that Cj-OMVs possess proteolytic activity associated with the three proteases HtrA, Cj0511, and Cj1365. Among them, the HtrA and Cj1365c proteases, have been shown to be responsible for the cleavage of the major AJs and TJs proteins E-cadherin and occludin, suggesting that the proteolytic activity of OMVs play an important role in enhancing the levels of *C. jejuni* interactions with intestinal epithelial cells [140]. Hp-OMVs, containing the cytotoxin-associated gene A (CagA), which plays a critical role in gastric inflammation and gastric cancer development, have been found to localize in the vicinity of cellular junctions causing the redistribution of the TJ protein ZO-1 in junctional areas. Furthermore, it has been observed that the CagA treatment of epithelial cells induces histone H1 to bind to ATP, probably with an effect on gene transcription, which may potentially lead to different outcomes of *H. pylori* infection and initiation of cancer [141].

On the other side, probiotic EcN, a good colonizer of the human gut with proven therapeutic efficacy in the remission of ulcerative colitis in humans, positively modulates the intestinal epithelial barrier through upregulation and redistribution of the tight junction proteins ZO-1, ZO-2, and claudin-14 [142]. These effects depended on OMVs released by EcN that can modulate the intestinal epithelial barrier together with a cytokine/chemokine response of gut epithelial and immune cells in in vitro and ex vivo cellular models [126]. In particular, EcN-OMVs increase the intestinal epithelial barrier through an up-regulated expression of secreted antimicrobial factors such as β-defensin-2 [126,143] and the TJ proteins ZO-1 [144], ZO-2 [145], and claudin-14 [146]. In addition, the effectiveness of this probiotic in the amelioration of induced experimental colitis in mice is well-documented [144,147,148,149,150]. Furthermore, it has been reported that EcN-OMVs are able to induce IL-22 expression in colonic explants [126]. IL-22, a cytokine that targets epithelial cells reinforcing the intestinal barrier limiting the access of microbial compounds and allergens to the systemic circulation, helps to preserve the integrity of the epithelial barrier by inducing mucin production by goblet cells [126]. Further studies have shown that the probiotic factor increasing the intestinal barrier function is the protein TcpC loaded into OMVs, which activates the ERK1/2 signaling pathway, resulting in the upregulation of the tight junction protein claudin-14 [146]. OMVs produced by other commensal *E. coli* strains, as reported for EcN-OMVs, are internalized by epithelial cells through clathrin-mediated endocytosis [151] and mediate signaling events to the immune system through the intestinal epithelial barrier [105]. Specifically, OMVs from commensal *E. coli* strains, ECOR63 and ECOR12, increase epithelial barrier function, upregulating the expression of ZO-1 and claudin-14 proteins and downregulating of the leaky protein claudin-2 by mechanisms not dependent on TcpC [152]. Similarly, OMVs produced by the intestinal commensal bacterium *Akkermansia muciniphila* (*A. muciniphila*), whose number reduction in gut microbiota has been linked with the development IBD [153,154], improved intestinal permeability by enhancing TJ function. In a high-fat diet-induced diabetic mouse model, *A. muciniphila*-derived OMVs (Am-OMVs) neutralized increased intestinal permeability in LPS-treated Caco-2 cells. Western blotting showed that vesicles upregulated TJ protein expression, including occludin, ZO-1, and claudin-5. The authors suggested that Am-OMVs could be used to positively modulate intestinal barrier integrity, reduce inflammation, and provide therapeutic potential for inflammatory disease [155]. To analyze whether internalized OMVs from commensal and probiotic *E. coli* elicit an inflammatory response through activation of NOD1 and/or NOD2 receptors, Caco-2 cells were transfected with siRNA-specific sequences targeting these NOD receptors. When transfected cells were stimulated with OMVs from EcN and ECOR12, NOD1 silencing and RIP2 inhibition significantly decreased expression of these pro-inflammatory IL-6 and IL-8 at both mRNA and protein levels [156]. These findings demonstrate that OMVs isolated from the probiotic EcN and the commensal ECOR12 activate NOD1 signaling pathways in intestinal epithelial cells [151]. The delivery of peptidoglycan inside the host cell and further activation of NOD signaling cascades by bacterial OMVs have been studied in pathogens, specifically *H. pylori* [106], *Vibrio cholerae* [114], and *Aggregatibacter actinomycetemcomitans* [157]. OMVs from these pathogens induce an inflammatory response that is NOD1-dependent, resulting in RIP2-dependent autophagy and inflammatory signaling in response to bacterial infection. In particular, the intracellular compartment where peptidoglycan contained in OMVs internalized via clathrin-mediated endocytosis interacts with cytosolic NOD1 was defined using Hp-derived OMVs. Results from this study revealed that such interaction takes place at the membrane of early endosomes, where NOD1 and RIP2 are recruited [158].

### 4.2. BEV Interaction with Lung Epithelium

The main barrier against the infiltration of microorganisms in the lung is the layer of epithelial cells that covers the surface of the alveoli and airways. It is known that pathogen bacteria use OMVs to invade the lung epithelial barrier without making direct contact with host cells, mediating inflammatory responses in vitro and in vivo [159]. During the infection of lungs, OMVs released by *Moraxella catarrhalis*, an emerging human respiratory pathogen in patients with chronic obstructive pulmonary disease (COPD) and in children with acute otitis media, and *Klebsiella pneuomoniae*, an important opportunistic pathogen causing various types of extraintestinal infections, have been shown to mediate pulmonary inflammation and induce neutrophil and lymphocyte infiltrations into the lungs [160,161]. In particular, activated neutrophils trigger oxidative stress and release proteases, resulting in lung damage [162]. Intraperitoneal administration of *E. coli*-derived OMVs induce neutrophil infiltrations into the lungs more potently than equivalent amounts of the purified LPS via TLR4 and endothelial intercellular adhesion molecule1-dependent manners [163]. Moreover, intranasal administration of *P. aeruginosa*-derived OMVs (Pa-OMVs) is able, without any live bacteria, to mediate pulmonary inflammation via TLR2 and TLR4 [164], suggesting that OMVs may be causative agents in the pathogenesis of specific infectious diseases. In particular, in bronchoalveolar lavage (BAL) at 24 h after their intranasal administration in mice, Pa-OMVs caused a significant increase in the neutrophils and macrophages, and the concentrations of the chemokines CXCL1 and CCL2, involved in the chemotaxis of neutrophils and macrophages, and of the cytokines IL-1β, TNF-α, IL-6, and IFN-γ were, found to be dose-dependent [164].

It has been demonstrated that there are interesting links between BEVs with allergic diseases, such as asthma and atopic dermatitis [165,166]. Asthma is a chronic inflammatory disease characterized by inflammation in the airways and airway hyper-responsiveness against inhaled allergens [167]. It has been demonstrated that the biological ultrafine particles derived mainly from bacteria, including LPS and BEVs [168], are related to the pathogenesis of chronic inflammatory pulmonary diseases, such as asthma. BEVs, contained in indoor dust, are secreted by Gram-negative and Gram-positive bacteria. In this regard, it has been found that dust BEVs derived from Gram-negative bacteria can induce neutrophilic pulmonary inflammation, which is associated with Th1 and Th17 cell responses. Particularly, Gram-negative bacteria-derived OMVs were isolated by indoor dust collected from a bed mattress in an apartment through 0.22 μm filtration and centrifugation to remove bacteria and then ultrafiltration to concentrate them. After dynamic light scattering and electron microscopy analysis and LPS detection, dust OMVs were applied intranasally to C57BL/6 mice twice per week for 4 weeks, causing lung infiltration of inflammatory cells. In particular, BAL analysis showed that labeled dust OMVs were internalized by alveolar macrophages and lung epithelial cells, increasing the production of TNF-α and IL-6 in a dose-dependent manner compared with inhalation of PBS as a control. In addition, the polimixin B treatment abolished the pulmonary inflammation induced by dust OMVs, suggesting that Gram-negative-bacteria-derived BEVs in indoor dust are crucial in the development of pulmonary inflammation [166]. Many studies have shown that *Staphylococcus aureus* (*S. aureus*) and *S. aureus* BEVs are also present in indoor dust, indicating that this bacterium is able to produce BEVs in dust [169,170]. Different doses of *S. aureus* BEVs administered to mouse airways were internalized by alveolar macrophages and airway epithelial cells, inducing distinct pro-inflammatory mediator profiles by these kinds of cells. Histological analysis of lung tissues also showed that an increase in the inflammatory cell infiltration was enhanced by exposure to *S. aureus* BEVs along with Th1 and Th17 cell response and neutrophilic pulmonary inflammation induced, suggesting that *S. aureus* BEVs in indoor environments may be an important causative agent of pulmonary inflammatory diseases.

### 4.3. BEV Interaction with Skin Epithelium

The epidermis is the outermost layer of the skin, consisting of a stratified squamous epithelium that is composed mainly of keratinocytes in various stages of differentiation. Epidermal keratinocytes are highly specialized epithelial cells that play an essential role in providing skin structure and in the functioning of the immune system [171]. The disturbance of the skin microbial balance can result in dysbiosis-promoting skin diseases, such as atopic dermatitis, in addition to host genetic susceptibility. Specifically, *S. aureus* plays a crucial role in atopic dermatitis development, for being found in almost all lesional skin in atopic dermatitis patients, whereas a reduction in its colonization has been shown to decrease disease severity [165,172]. Moreover, it has been demonstrated through in vitro and in vivo studies that *S. aureus* BEVs are also causative agents of atopic dermatitis *S. aureus* BEVs, ranging in size between 20 and 130 nm, contain more than 100 proteins, including extracellular and membrane-associated virulence factors such as toxins, adhesins, proteolysin, coagulase, and other related enzymes, which play as well as virulence-associated molecules [173]. Furthermore, *S. aureus* BEVs have been found to induce apoptotic cell death in a dose-dependent manner when HEp-2 cells, human epithelial cells derived from a larynx carcinoma, were treated with them [174], thereby demonstrating the cytotoxic properties of *S. aureus* BEVs. On the contrary, BEVs obtained from cultured *S. aureus* isolated from atopic dermatitis lesions are not cytotoxic to keratinocytes [175], suggesting that this discrepancy in their cytotoxicity may be due to differences in *S. aureus* strains or in cell lines used in these studies. Immunohistochemical analysis performed in skin biopsy samples obtained from atopic dermatitis lesions showed the presence of staphylococcal protein A (SPA), used as a marker of *S. aureus* BEVs for the large content of this protein within them [175]. Moreover, by immunoelectron microscopic analysis, SPA was detected in the cytoplasm of keratinocytes as well as in the intercellular space of the epidermis of atopic dermatitis lesions, suggesting the ability of *S. aureus* to deliver BEVs carrying effector molecules to the epidermis in atopic dermatitis lesions. When HaCaT cells, the immortalized human keratinocytes used to study the epidermal homeostasis and its pathophysiology [176], were treated with *S. aureus* BEVs, the expression of pro-inflammatory cytokine genes, including IL-1b, IL-6, IL-8, and MIP-1a, was increased compared to the controls. The gene expression of these of pro-inflammatory cytokines was not in a dose-dependent manner, confirming that keratinocytes are the main producers of cytokines and chemokines in the skin, which are necessary for inducing immune responses against pathogens. It has been established that peptidoglycan in the *S. aureus* BEV surface is likely responsible for inducing pro-inflammatory responses mediated by the TLR2 and NOD2 signaling pathways, as observed by experiments of interfering RNA transfection to induce TLR2, TLR6, NOD1, or NOD2 gene silencing [175].

### 4.4. BEV Interaction with Oral Epithelium

The gingival epithelium is the first line of defense in the oral cavity [177]. It has been found that *P. gingivalis*, a predominant Gram-negative periodontal pathogen, secretes OMVs (Pg-OMVs) involved in the etiology of periodontitis, which is initiated through dysbiosis of oral microbiota. In the context of periodontitis, characterized by gingival inflammation, as well as a loss of connective tissue and bone from around the roots of the teeth, Pg-OMVs can be considered as potent vehicles for the transmission of virulence factors [178,179]. Several studies using the proteomic approach have shown that Pg-OMVs contain various virulence factors, including fimbriae, mediating bacterial cell, and OMV adherence to and entry into periodontal cells through lipid raft-dependent endocytic pathways, proteases termed gingipains, contributing to the destruction of periodontal tissues [178,179,180]. After Pg-OMVs entry, OMV-associated gingipains degrades the cellular transferrin receptor (TfR), indispensable for iron metabolism, and integrin-related signaling molecules, such as paxillin and focal adhesion kinase (FAK), which result in depletion of intracellular transferrin and inhibition of cellular migration [181]. These molecules control cellular migration and proliferation, which are crucial functions for wound healing and tissue regeneration of periodontal tissues destroyed during periodontitis. Furthermore, Pg-OMVs are able to induce a strong inflammatory response when they are added to a culture of immortalized human gingival epithelial (HGE) cells [182]. In particular, using real-time RT-PCR, HGE cells, stimulated by Pg-OMVs, increased mRNA expression levels of cyclooxygenase (COX-2), IL-6 and IL-8, and matrix metalloproteinase (MMP)-1 and MMP-3. During periodontitis, it has been established that PGE2 produced by COX-2, is a potent stimulator of bone resorption together with IL-6, whereas IL-8 leads to recruitment and infiltration of neutrophils, and MMP-1 and MMP-3 are responsible for the excessive breakdown of periodontal tissue. These cellular inflammatory responses induced by Pg-OMVs were found to be suppressed by the action of hop bract polyphenol [182]. The role of BEVs in viral infection constitutes an emerging area of research still to be explored. It has been found that, by in situ PCR, HIV-1 was detected within oral mucosal epithelial cells in a significant majority of infected subjects [183]. It has been demonstrated that HIV-1 bound to the Pg-OMVs, through the interaction between the binding domains of gingipains and HIV-1 gp120, allowing the virus to enter non-permissive epithelial cells via vesicle endocytosis. Once inside the cell, the virus establishes infection, as evidenced by successful integration of the virus into the epithelial genome, although the mechanism by which HIV-1 dissociates from Pg-OMVs and is released into the cytoplasm of cells is currently not known [184,185].

## 5. BEV Interaction with Immune System

The immune system plays a crucial role in keeping the body healthy by providing a fine balance between the elimination of invading pathogens and the maintenance of tolerance to healthy self-tissue. The crosstalk between commensal bacteria and the immune system is crucial in establishing and maintaining immune homeostasis and preventing immune-mediated disorders such as autoimmunity, allergies, and chronic inflammatory diseases [186,187]. Current evidence from animal models indicates that a bidirectional relationship exists between microbiota perturbation and immune dysregulation. The cargo transported by BEVs can vary depending on a wide variety of factors including the bacterial species, bacterial strain, and growth phase [188]. OMVs carrying diverse assortment of proteins, lipoproteins, lipids, and nucleic acids reach the submucosa by transcytosis across M cells, specialized epithelial cells of the mucosa-associated lymphoid tissue [189]. Once in the lamina propria, translocated BEVs directly interact with resident gut immune cells, triggering suitable immune responses. The diverse immunomodulatory effects of BEVs depend largely on the specific parental bacterium. For instance, BEVs from pathogenic bacteria have the potential to exacerbate infection by dampening immune responses or trigger an excessive immune reaction resulting in sepsis. In contrast, BEVs from symbiotic or commensal bacterial species in the gastrointestinal tract confer protection promoting maturation and immunological tolerance [186].

Most of several families of pattern recognition receptors (PRRs) recognize pathogen-specific molecules such as LPS, flagellin, RNA, DNA, or peptidoglycan, known as microbe- or pathogen-associated molecular patterns (MAMPs or PAMPs) that are carried by BEVs. OMVs carry proteins, such as OMP FomA, one of the most expressed outer membrane proteins (OMPs) crucial for cell adhesion and with immunogenic properties, which is recognized by TLR2, flagellin, a subunit protein of the flagellum involved in bacterial motility recognized by TLR5, RNA recognized by TLR7 and TLR8, DNA recognized by TLR9, and peptidoglycan by NODs, as previously reported [190,191,192,193,194]. Even in the absence of gut barrier pathology, low levels of MAMPs associated with BEVs can be detected in the circulation of mice and humans, controlling myeloid and lymphoid cell production in the bone marrow and thymus, with regulatory effects on immune cell functions throughout the body [195,196]. Macrophages phagocytose the bacteria and BEVs and also secrete various products provoking downstream immune responses. On the other hand, macrophages can serve as host cells for certain pathogenic microorganisms. BEVs of these intracellular microorganisms interact with various components within the host cells in a complicated manner. The pro-inflammatory or anti-inflammatory effect seems to be relevant to specie types and stages of infection and sometimes bacteria-infected macrophages also release extracellular vesicles that contain pathogen-derived macromolecules affecting other immune cells [197].

The pro-inflammatory responses involve NOD1, NOD2, and several other nucleotide oligomerization domain-like receptors (NLRs). These NLRs function as microbial sensors within the innate immune system such as the NLR family known as NLR-pyrin domain containing 3 (NLRP3). The activation of NLRP3 by BEVs leads to the formation of an intracellular complex termed the inflammasome, a protein complex responsible for the maturation of caspase-1 and the subsequent secretion of mature IL-1β and IL-18. BEVs released by *Staphylococcus aureus* [198], *P. aeruginosa*, [199], *Bordetella pertussis* [200], *E. coli* [201], and *P. gingivalis* [202] can activate NLRP3 inflammasome in murine bone marrow-derived macrophages (BMDMs) or human macrophages (THP-1 cells and monocyte-derived macrophages), promoting the release of the mature cytokines IL-1β and IL-18 and the induction of pyroptosis, a form of cell death that is triggered by proinflammatory signals and associated with inflammation. BEVs can modulate the immune response mediated by macrophages through a switch of their pro-inflammatory phenotypes to anti-inflammatory phenotypes. For example, Hp-OMVs induce IL-10 production in human PBMCs [203], while OMVs from *P. gingivalis* facilitate a loss of CD14 expression on macrophages [204], rendering these cells unresponsive to TLR4 signaling and effectively avoiding hyperinflammatory immune responses. Macrophage inflammasome activation was also shown to be induced by OMVs via different pathways.

OMVs released by *Neisseria gonorrhoeae*, uropathogenic *E. coli*, and *P. aeruginosa* have been shown to induce mitochondrial apoptosis and NLRP3 inflammasome activation [205]. Flagellated bacteria, such as *S. typhimurium* and *P. aeruginosa*, released OMVs containing flagellin that, in addition to TLR5, can trigger NLRC4 canonical inflammasome activation via flagellin delivery to the cytoplasm of host cells, inducing caspase-1 activation and interleukin-1β secretion [191]. All of these observations suggest that different BEVs containing different cargos may take different routes to mediate inflammation via the inflammasome, highlighting the complexity of BEV-mediated immune responses. OMVs from *P. gingivalis* were shown to reduce TNF released by U937 macrophage-like cells following stimulation with *E. coli* LPS. This regulation was attributed to arg-gingipain from *P. gingivalis*. Indeed, this proteolytic enzyme could hydrolyze the LPS co-receptor CD14 at the surface of macrophages, limiting the response to the endotoxin [204], contributing to the pathogenicity of *P. gingivalis* and other causative agents of periodontal diseases. Conversely, BEVs from *Pediococcus pentosaceus* have demonstrated potent anti-inflammatory properties, facilitating the differentiation of bone marrow precursors into myeloid-derived suppressor-like cells and promoting M2 macrophage polarization, most likely in a TLR2-dependent manner, both in in vitro and in vivo experiments [198,206]. Interestingly, a recent study showed that an outer membrane porin protein (PorB) carried by OMVs produced by *Neisseria gonorrhoeae* can cause apoptotic cell death and loss of mitochondria membrane potential in ex vivo macrophages. This finding raises the possibility that the strategy of targeting mitochondria and interfering with mitochondrial functions might be a conserved virulence mechanism across bacteria [207]. *S. aureus*-derived BEVs contain sRNA and DNA that have been implicated in inducing IFN-β mRNA by the activation of Toll-like receptors in mouse macrophages [208]. In a recent study aiming to investigate the effect of RNAs within *Aggregatibacter actinomycetemcomitans*-derived OMV (Aa-OMVs) on TNF-alpha production by the macrophage cells, U937 cells were treated with intact Aa-OMV and nuclease-treated Aa-OMV lysate [209]. The results indicated a significant reduction in TNF-α release and transcript levels when U937 cells were treated with nuclease-treated OMV lysate but not with OMV lysate without nuclease treatment, suggesting that sRNAs in Aa-OMVs stimulate TNF-α production via TLR8 and NF-κB signaling pathways. Additional experiments were conducted to determine which sRNS in OMVs are responsible for the TNF-α signaling pathways by using the RIP-Seq approach to determine whether sRNAs in Aa-OMVs can bind to the host RNA-induced silencing complex, which contains Ago2, thereby functioning as miRNAs. For Ago2 RNA immunoprecipitation sequencing (RIP-Seq) experiments, after Aa-OMVs incubation of U937 cells, total RNA bound to Ago2 were sequenced. Next-generation sequencing of Ago2-associated sRNAs yielded 24.9 million reads; among them, 0.35% were of *A. actinomycetemcomitans*, which constitute novel sRNAs [209]. This finding suggests that sRNAs carried by OMVs in addition to having the characteristic functions of exogenous miRNAs can act in a novel way as host regulators. In the immunological response, dendritic cells (DCs) play a key role in connecting the innate and adaptive immune systems [210]. Bacteria and BEVs have been shown to be internalized into DCs up-regulating CD86 and MHCII molecules and producing TNF and IL-12 [211].

*Bacteroides fragilis* (*B. fragilis*) produces a capsular polysaccharide (PSA), an immunomodulatory molecule that induces regulatory T cells and mucosal tolerance [212]. In particular, it has been demonstrated that DCs, in detecting PSA found in *B. fragilis*-derived OMVs (Bf-OMVs) via TLR2, activated Growth Arrest and DNA-Damage-Inducible protein (Gadd45α), resulting in anti-inflammatory cytokine IL-10 secretion and in increased proliferation of Tregs [127]. Furthermore, mutations in NOD2 and ATG16L1, genes associated with IBD and with ATG16L1 coding protein as a component of a large protein complex essential for autophagy, disrupt DC-Treg cell interactions, thereby blocking the protective function of Bf-OMVs [126]. Additionally, *E. coli*-derived OMVs (Ec-OMVs) induce DCs to generate T-helper cell responses in a strain-specific manner. The probiotic EcN and the commensal ECOR63 trigger increased secretion of Th1 polarizing cytokines, IFN-γ and IL-12, from DCs. Conversely, OMVs released by commensal ECOR12 stimulate the production of higher levels of IL-10 and TGF-β, cytokines related to Tregs and low levels of proinflammatory cytokines [213]. These studies indicated that BEVs released by gut microbiota strains stimulate the innate immune system through the activation of DCs, which, in turn, orchestrate balanced immune tolerance and immune inflammatory responses that are crucial for maintaining intestinal homeostasis.

Rheumatod Arthritis (RA) patients are more prone to develop periodontal diseases [31]. The main kind of autoantibodies highly specific to RA patients are anti-citrullinated protein antibodies [214]. Interestingly, an enzyme, peptidyl-arginine deiminase (PPAD), involved in the citrullination of bacterial and host proteins is associated with OMVs, together with several other virulence factors of the bacterium [179,215]. This enzyme may have a profound impact on the development and progression of RA via citrullination of proteins to generate neo-epitopes [216,217], although this remains under investigation. It has been suggested that bacterial and host proteins citrullinated by PPAD might initiate the loss of tolerance to citrullinated autoantigens in RA [216]. OMVs are enriched with LPS and outer membrane proteins known as potent immunostimulators [50,218]. OMVs have been shown to initiate a robust systemic inflammatory response that causes damage to end organs and even death in animal models [219]. In animal models, it has been demonstrated that OMVs derived from intestinal *E. coli* are causative microbial signals in the pathogenesis of systemic inflammatory response syndrome (SIRS) and sepsis-induced lethality through the systemic induction of TNF-α and IL-6 [219]. Furthermore, it has been shown that *Neisseria meningitidis* released OMVs in the plasma of patients with severe sepsis [220].

Sepsis is a syndrome arising from an excess of the host immune response to an infection, which leads to multiple organ failure and systemic inflammation. The inflammatory system becomes hyperactive, which involves infiltration of inflammatory cells and increased production of proinflammatory mediators such as TNF-α and IL-6 [221]. Moreover, the coagulation system is triggered through extreme activation of platelets, which provokes disseminated intravascular coagulopathy [222]. Using a mouse model, the administration by intraperitoneal injection of high doses of Ec-OMVs (25 and 50 µg) induced a strong systemic inflammatory response leading to death within 24 h after injection [219]. The inflammation triggered by intraperitoneal injection of Ec-OMVs induced systemic inflammatory response syndrome including the up-regulation of leukocyte infiltration in BAL fluid, increase in pro-inflammatory cytokines, and increased lung tissue permeability [219]. Hypotension or disseminated intravascular coagulation has been related to other noteworthy characteristics of severe sepsis [223,224,225]. OMV treatment resulted in a reduction in platelets in peripheral blood and an increase in D-dimer levels in plasma, showing that OMVs produced disseminated intravascular coagulation [226]. In particular, wild-type or TLR4 knock-out mice were intraperitoneally injected with Ec-OMVs or with *E. coli* with genetic deletion of ypjA gene, which is critical for OMV production. Genetic deletion of ypjA significantly attenuated E.coli-induced coagulopathy, intravascular thrombi deposition, multiple organ injuries, and mortality, whereas the injection of purified Ec-OMVs resulted in the development of disseminated intravascular coagulation in a TLR4-dependent manner [226]. Since severe sepsis is accompanied by systemic inflammation that results from excessive release of cytokines into the systemic circulation [227,228], it has been observed that the serum levels of TNF-α and IL-6 were markedly enhanced 6 h after multiple injections of 5 µg Ec-OMVs in mice. These findings suggest that OMVs might be efficient enough to induce systemic inflammation in distant organs, although bacteria themselves are important for the development of sepsis at the first infection site.

As commensal bacteria can alter infection of enteric viral pathogens, studies have investigated the ability of OMVs derived from commensal Gram-negative bacteria, such as *Enterobacter cloacae* and *Bacteroides thetaiotaomicron*, to modulate the innate immune responses against the norovirus for controlling the viral replication [229,230,231]. Noroviruses, a small, non-enveloped, single-stranded, positive-sense RNA virus causing gastroenteritis in humans and spreading to the fecal-oral route, is able to infect macrophage-like cells in vivo and replicate in cultured primary DCs and macrophages [232]. Using murine norovirus (MNV) as a surrogate to explore human norovirus pathogenesis, it has been shown that CD36, a scavenger receptor [233], and CD44, a cell surface receptor for the extracellular matrix ligand hyaluronate, are involved in MNV-1 binding to primary DCs, while CD98 heavy chain (CD98), a multifunctional glycoprotein, and transferrin receptor 1 (TfRc), are involved in MNV-1 binding to RAW 264.7 cells, a murine macrophage cell line [234]. Furthermore, it has been observed that there is a significant increase in the BEV released from gut microbiota during MNV-1 infection [235]. It has been found that MNV is able to bind to OMVs from *Enterobacter cloacae* and *B. thetaiotaomicron*, thereby facilitating both virus and OMV uptake by target cells, such as RAW264.7 cells [231]. Specifically, the MNV-OMV complex entering macrophages induces higher production of pro-inflammatory cytokines, among which IL-1β, TNF-α, and IFN-β, constitute the specific antiviral immune pathways. Given that MNV infection increases BEV production in vivo, these findings suggest that OMVs from *E. cloacae* and *B. thetaiotaomicron* are able to increase innate antiviral immune responses, which would lead to the control of MNV infection.

## 6. BEV Interaction with CNS

Increasing evidence suggests that gut and oral dysbiosis constitute a key marker of neuronal disorders such as multiple sclerosis and neurodegenerative diseases, such as Parkinson’s and Alzheimer’s diseases (AD) [236]. How the alterations in the gut/oral microbiota relate to the pathological events seen in the brain is yet unknown, although few mechanisms for brain–gut microbiota crosstalk have been proposed, including modified hypothalamic–pituitary–adrenal responses [237], immune system activation [238], and vagus nerve activation [239]. Very recent studies have suggested that, in addition the bidirectional interaction between the gut/oral microbiota and the CNS, a biological pathway of communication between the lungs and the brain may exist [240,241]. In particular, studies on experimental autoimmune encephalomyelitis (EAE), an animal model of multiple sclerosis (MS) [242], have established that perturbing the lung microbiota with the antibiotic neomycin, affecting the composition of the lung microbiota towards a higher abundance of Gram-negative bacteria producing LPS, resulted in the amelioration of EAE [243,244]. Indeed, LPS, whose increased levels were detected in the BAL of neomycin-treated rats, crossing the BBB, can induce microglia to produce I IFN, which is crucial to limit EAE progression [245]. Given that OMVs carry LPS and can cross BBB to enter CNS, it would be interesting to focus on the potential pathogenic role of circulating BEVs in EAE and MS, as to date no research in this field has been conducted.

Recently, since BEVs secreted by the gut and oral microbiota have been observed to possess the ability to penetrate the BBB, by mechanisms not fully understood, BEVs have been postulated to be one of the key mechanisms contributing to the gut–brain communication. Therefore, research into the potential role of BEVs in crossing the BBB and causing neuroinflammation is still an area of active investigation. A few studies have also reported the presence of bacterial nucleic acids in the brain [97,98]. Furthermore, research has detected the presence of OMVs released by *P. gingivalis*, the keystone pathogen in chronic periodontitis, in the peripheral blood and cerebrospinal fluid (CSF) in animal models with severe bacterial infections [245]. In light of these findings and in view of the presence of BEVs in the bloodstream and their capability to cross the epithelial barrier, it is possible to hypothesize that a fraction of the circulating BEVs might gain access to the brain through the BBB (Figure 4). It has also been hypothesized that [84] the most likely mechanisms of BEV distribution to the CNS are by crossing the BBB [246] and vagal nerve transport [247]. However, the recently discovered meningeal lymphatic vessels reveal a route to the CSF–brain barrier not yet explored and that BEVs could be utilized to access the brain parenchyma [248]. The ability of BEVs to cross the BBB and to induce neuroinflammation and cognitive impairment has been studied in vivo using different murine models [194,249,250,251,252,253,254], employing OMVs labeled with lipophilic dyes that were administrated in mice by oral gavage, or as an injection to blood circulation, either by intracardiac injection or through the tail vein.

In a recent study, it was demonstrated that BEVs derived from the feces of AD patients are able to enter the brain, breaking the BBB and inducing marked neuroinflammation with specifically tau hyperphosphorylation aggravation [249]. Indeed, the pathological hallmarks of AD are extracellular accumulation of amyloid-β (Aβ) plaques composed of Aβ peptides and neurofibrillary tangles (NFTs) composed of a highly-phosphorylated form of the microtubule-associated protein tau [255]. Furthermore, Glycogen synthase kinase-3 (GSK3) is involved in various cell biology pathways, some of which have also been implicated in neurodegeneration [256] and found hyperactive in the brain of AD patients [257]. In this OMV transplantation study, BEVs purified from feces of AD patients (AD-BEVs) and from healthy controls (HC-BEVs) were labeled with PKH26 and injected via the tail vein in mice. The results showed the presence of labeled-AD-BEVs in the hippocampus by fluorescence microscopy and the concentration of Evans blue dye, used to test BBB permeability, significantly higher in the group of mice AD-BEV-injected than in the group of mice HC-BEV-injected. These findings suggest that AD-BEVs entered the brain, affecting the BBB permeability. A similar evidence was also provided by Western blot analysis for claudin-5, an important protein for the tight junctions, whose expression level showed a significant decrease in mice AD-BEV-injected than mice HC-BEV-injected. Furthermore, in mice, AD-BEVs induced cognitive impairment as tested by Morris water maze (MWM) for detecting spatial learning and memory ability, and an increase in GSK-3β activity and tau hyperphosphorylation as detected by Western blot analysis [249]. To explore the potential role of Pg-OMVs to induce neuroinflammation in the hippocampus and behavioral cognitive changes, Pg-OMVs, purified from bacteria cultures, were labeled with DiO and administrated to mice by oral gavage. Labeled Pg-OMVs were clearly detected in the hippocampus and cortex. The Pg-OMVs entry to the brain induced BBB alteration as indicated by Western blot analysis and RT-qPCR to detect tight junction–related protein and gene expression, respectively. In particular, RT-qPCR showed that claudin-5, ZO-1, and occludin gene expression were decreased in the hippocampus of Pg-OMV treated mice, meanwhile the Western blot indicated that occludin was significantly decreased in the hippocampus of Pg-OMV treated mice. Moreover, Morris water maze and Y-maze tests demonstrated that Pg-OMV treated mice showed learning and memory impairment [251]. Furthermore, oral gavage of Pg-OMVs increases tau phosphorylation and activated the NLRP3 inflammasome in the hippocampus of mice. All of these findings indicate that chronic infection by oral gavage of Pg-OMVs induced AD-like phenotypes, including learning and memory deficiency, microglia-mediated neuroinflammation, and intracellular tau phosphorylation. Since *Paenalcaligenes hominis* (*P. hominis*), a Gram-negative bacterium, which has been found to cause strong cognitive impairment and colitis in germ-free mice [258], and *E. coli*, were both frequently detected in the feces of aged mouse or elderly adults but not in children, young adults, and young mice [259]. Studies have explored the effects of Ec-OMVs and *P. hominis*-derived OMVs (Ph-OMVs) on the occurrence of age-related degenerative cognitive impairment and colon inflammation in mice [247]. As the transplantation of feces from elderly people or aged mice, containing larger populations of *P. hominis* and *E. coli* than young adults or young mice, caused cognitive impairment and colitis in transplanted young mice, so also oral gavage of Ph-OMVs caused cognitive impairment and colitis in mice. In particular, it has been demonstrated that after oral gavage, Ph-OMVs conjugated with FITC more strongly accumulated in the hippocampus, but vagotomy, an intervention to interrupt signals carried by vagal nerve, reduced their translocation into the brain. These findings suggest that Ph-OMVs may enter the brain across BBB from the blood or vagus nerve, suggesting that OMV interaction with the vagus nerve is at least partially responsible for changes to the brain [247].

Numerous studies have shown that the infection of *H. pylori* has been found to be associated with an increased risk of developing AD [260,261]. The effects of Hp-OMVs on brain functions and AD pathology have recently been studied in wild-type (WT) mice and App^NL-G-F^ AD mice, a second-generation mouse model of AD in which, unlike most other transgenic mouse models, the mutant amyloid precursor protein (APP) is not overexpressed, thereby avoiding potential artifacts [262]. Hp-OMVs purified from bacterial cultures by a combination of ultrafiltration and size exclusion chromatography (SEC) were administered to WT and App^NL-G-F^ AD mice by oral gavage. After the administration of Hp-OMVs in WT mice, the integrity of gastrointestinal barrier, BBB, and blood-CSF barrier, analyzed using 4 kDa FITC-conjugated dextran tracer, was maintained. In agreement with this finding, differences in mRNA levels of the TJs in hippocampus and choroid plexus upon Hp-OMV treatment were not observed. When Hp-OMVs were administrated to App^NL-G-F^ AD mice by oral gavage, after 3 weeks, Aβ plaque formation was detected. Once in the brain, Hp-OMVs are taken up by astrocytes, which induce the activation of glial cells and neuronal dysfunction, ultimately leading to exacerbated amyloid-β pathology and cognitive decline [263]. Furthermore, the Hp-OMV-treated App^NL-G-F^ AD mice showed cognitive impairment based on the Y-maze task and performance to the new object in the NOR test, measuring the tendency of mice to search for new objects compared to old ones. Strong evidence implicates the complement pathway as an important contributor to amyloid pathology in AD, showing that the expression of C3 and C3a receptor (C3aR1) is positively correlated with cognitive decline. Otherwise, the deletion of C3aR1 in mice results in the rescue of tau pathology and attenuation of neuroinflammation, synaptic deficits, and neurodegeneration [263,264]. Interestingly, the treatment with C3aR1 antagonist (C3aRA) prevented the occurrence of cognitive impairment caused by Hp-OMVs in App^NL-G-F^ AD mice. These findings provide new insights supporting the infectious hypothesis behind AD pathogenesis and also identify OMVs as important players in the gut–brain axis [254].

The majority of RNAs in OMVs are sRNAs [265,266], which might have a regulator role in host cells [81,266,267,268]. Although their precise mechanism of action has not been clearly elucidated, bacterial sRNAs are known to have specific roles in host gene regulation, immune response, and diseases [269]. Recently, in mice, the intracardiac injection Aa-OMVs labeled with DiD dye and/or Syto RNA-Select showed the Aa-OMV ability to cross BBB in a dose-dependent manner and their sRNA cargos to increase TNF-α expression in the mouse brain [209]. In particular, Aa-OMVs crossing mouse BBB were present in the brain 24 h after intracardiac injection, inducing the production of proinflammatory cytokine TNF-α, which has been found to be elevated in the serum of patients with AD [270]. However, it was not clear what type of brain cells were responsible for the TNF-alpha production caused by Aa-OMV carrying sRNAs [271]. In another study aiming to explore whether Aa-OMVs can be taken up by mouse microglial cells, Aa-OMVs stained with SYTO RNA Select Green were injected through the tail vein of CX3CR1-GFP transgenic mice, and their biodistribution using intravital imaging technique was analyzed [194]. The G-protein-coupled receptor CX3CR1 is expressed in human monocytes, and CX3CR1-GFP transgenic mice are widely used in studies on microglial cells that report a high density of meningeal macrophages in the dura mater and pia mater [272,273]. Along with other CNS macrophages (perivascular and choroid plexus), meningeal macrophages have been reported to be non-parenchymal immune modulators at brain boundaries. For the first time, studies have demonstrated the transport of Aa-OMVs from peripheral to brain microglial cells through the meninges, as observed by colocalization of Aa-OMVs and microglial cells using intravital imaging analysis, suggesting that Aa-OMVs and RNA cargo can enter microglial cells [194]. Furthermore, to explore the Aa-OMVs-sRNA effect on microglia cells, murine BV2 microglial cells were treated with Aa-OMVs and OMV lysate. Interestingly, the level of IL-6 was upregulated by intact OMVs but not by OMV lysates treated with nucleases, indicating that the nucleic acid cargo of Aa-OMVs was responsible for the activation of IL-6 [194]. The overall findings indicated that sRNAs within Aa-OMVs promote the production of IL-6 in microglial BV2 cells through NF-κB activation via OMVs.

A study verified the hypothesis that periodontitis can promote the migration of oral BEVs to the brain, inducing neuroinflammation. For this study, a ligature-induced periodontitis model mouse and OMVs produced by *A. actinomycetemcomitans*, a Gram-negative bacterium often found in association with localized aggressive periodontitis, were used [274]. After intragingival injection of DiD-labeled Aa OMVs, the fluorescence signals of Aa OMVs were observed exclusively in maxillary or V2 trigeminal ganglion (TG) neurons, suggesting that Aa OMVs enter the TG neuronal cell body through retrograde axonal transport, where they directly trigger proinflammatory neuronal signals. The mechanism by which Aa OMVs can be transported to the neuronal soma through the axon terminal might be the retrograde signaling by “signaling endosomes from distal axons to cell bodies and/or dendrites”, a well-known transmission process for some neurotransmitters [275]. In addition to oral BEVs’ ability to reach CNS through bloodstream and BBB crossing, these findings suggest a novel potential route by which oral BEVs associated with periodontitis can induce neuroinflammation. In particular, BEVs can also enter the trigeminal ganglion neuronal cell body through retrograde axonal transport, where they directly induce proinflammatory mediator expression.

**Table 1 ijms-25-08722-t001:** List of bacterial species producing BEVs reported in the text.

Bacterial Species	References	Gram Stain
*Aggregatibacter actinomycetemcomitans*	[157,194,209,274,275]	Gram negative
*Akkermansia muciniphila*	[153,154,155]	Gram negative
*Bacteroides fragilis*	[127]	Gram negative
*Bacteroides thetaiotaomicron*	[88,128,231]	Gram negative
*Bordetella pertussis*	[200]	Gram negative
*Campylobacter jejuni*	[140]	Gram negative
*E. coli Nissle 1917*	[126,151,152,213]	Gram negative
*Enterobacter cloacae*	[231]	Gram negative
*Enterohemorrhagic E. coli*	[91,193]	Gram negative
*Enterotoxigenic E. coli*	[109,123]	Gram negative
*Escherichia coli*	[82,126,138,156,163,205,213,219,226,267,268]	Gram negative
*Fusobacterium nucleatum*	[132]	Gram negative
*Helicobacter pylori*	[51,52,53,54,55,56,57,58,59,60,61,63,64,74,80,89,112,141,203,253,254]	Gram negative
*Klebsiella pneuomoniae*	[161]	Gram negative
*Moraxella catarrhalis*	[160]	Gram negative
*Neisseria gonorrhoeae*	[205,207]	Gram negative
*Neisseria meningitidis*	[218,220]	Gram negative
*Paenalcaligenes hominis*	[247]	Gram negative
*Pediococcus pentosaceus*	[206]	Gram positive
*Porphyromonas gingivalis*	[72,90,122,178,179,181,182,184,204,215,217,245,251,252]	Gram negative
*Pseudomonas aeruginosa*	[66,68,72,110,113,164,199,205]	Gram negative
*Salmonella typhimurim*	[72,191]	Gram negative
*Staphylococcus aureus*	[173,174,175,188,198,208]	Gram positive
*Vibrio cholera*	[114]	Gram negative

## 7. Conclusions

Gut and oral microbiota-derived BEVs are currently receiving a growing interest for their ability to cross epithelial barriers and enter systemic circulation to transport and deliver their bioactive molecules into target cells all over the body. BEV research has been revealing mechanisms by which commensal and pathogenic bacteria interact with host cells, modulating host signaling pathways and cell processes, leading to anti-inflammatory/pro-inflammatory responses. The knowledge about mechanisms by which BEVs contribute to various physiological and pathological processes is extremely complex due to their multicomponent cargo of active molecules, which can undergo compositional changes in response to different physiological and pathological conditions. Therefore, clarifying the composition of BEVs is crucial for understanding their functions.

The metagenomic profile of BEVs detected in blood and urine samples of patients and healthy individuals highly correlates with gut microbiota, stimulating new areas of investigation in microbiota research and garnering interest in BEVs as potential diagnostic markers. The effects of BEVs on the microbiota–gut–brain and microbiota–oral–brain axes and their potential function in neurodegenerative diseases have been explored in animal models. Despite the fact that BEVs have been characterized to cross BBB in animal models and in vitro cell cultures, the mechanism by which BEVs enter the brain remains poorly characterized. However, it has been shown that BEVs containing harmful molecule, such as LPS, peptidoglycans, toxins, and nucleic acids, can reach the brain by crossing BBB through the vagus nerve or in the context of periodontitis through a trigeminal neuron-mediated trans-neural pathway, leading to neuroinflammation and cognitive impairment. Moreover, the recently discovered meningeal lymphatic vessels can induce to investigate them as a novel BEV route to access the brain parenchyma not yet explored.

Based on their properties as long-distance delivery vehicles, stability in transport and storage, immunomodulatory properties, and ease of engineering, BEVs are being developed for clinical applications and explored as promising novel therapies, such as new vaccines and adjuvants or specialized drug delivery tools for the treatment of various diseases. Increasing evidence suggests that BEVs may have significant potential as diagnostic biomarkers for various bacterial infections, since they contain biomolecules, including DNA, allowing us to identify the specific type of bacteria responsible for the illness. Therefore, BEV-metagenomics and immunoassays can be used to develop a novel diagnostic methodology. However, the exploration of BEVs as biomarkers in disease is still in its early stage. BEVs from commensal bacteria, having immunomodulatory properties, can be used to combat virus infection by activating the host immune system. Commensal BEVs can trigger antiviral immune responses through a type I IFN-dependent mechanism, blocking viral replication. Research in the field of BEVs is rapidly developing and the BEV-based platform has been showing great promise for biomedical applications.

## Figures and Tables

**Figure 1 ijms-25-08722-f001:**
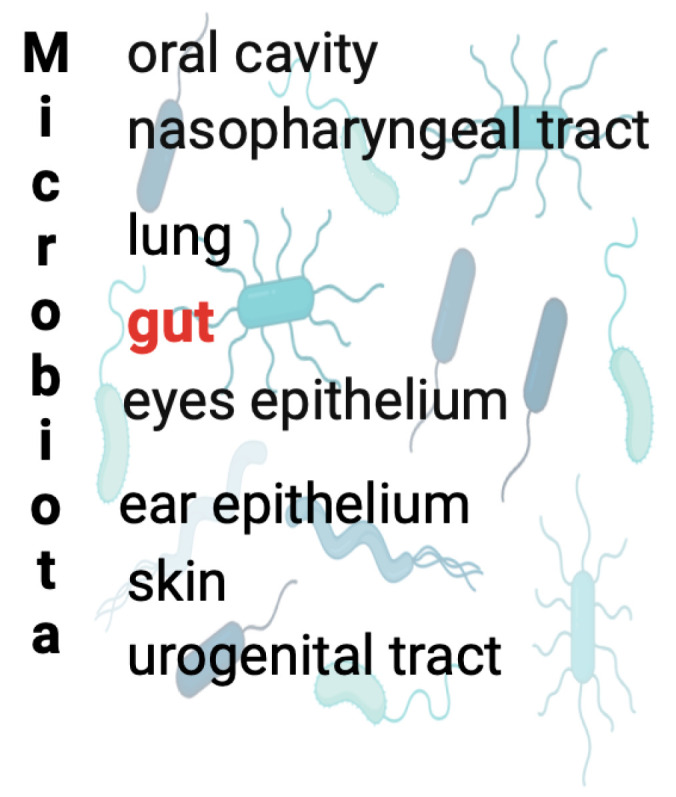
Schematic representation of microbiota’s presence in the human body. In red is indicated the gut microbiota that represents more than 99% of the total microbial community within the body (created with BioRender.com).

**Figure 2 ijms-25-08722-f002:**
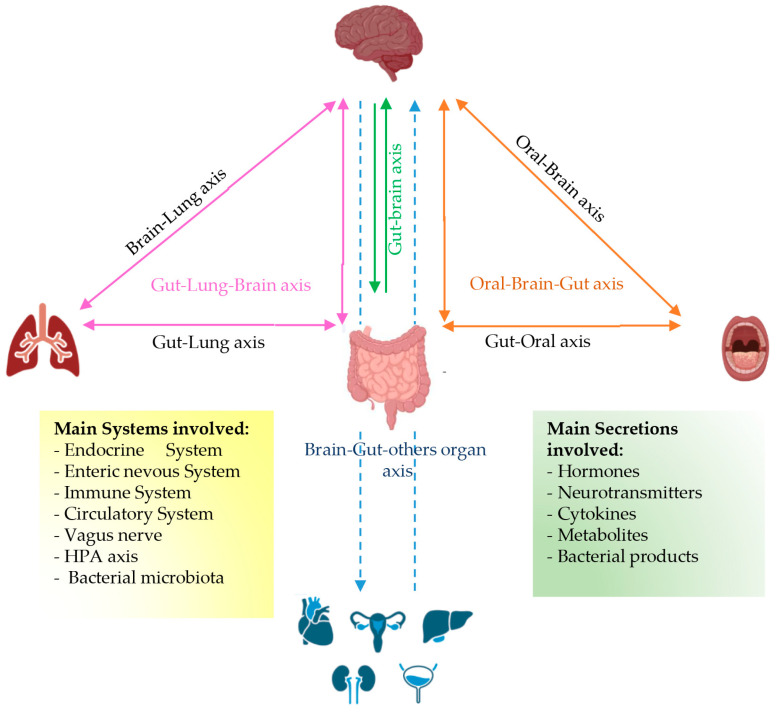
Schematic representation of communication network in the body. The communication network in the body is a complex and interconnected system that allows for different organs and systems to work together. There are different axes in the human body that represent bidirectional or multi-directional communications among different body compartments consisting not only of anatomical connections but also of molecules derived from the immune and endocrine systems, metabolites transported through the bloodstream, and bacterial products including BEVs originating from microbiota residing in the different organs. The gut is the key place of interaction with other organs. The brain–gut axis (green) is a complex connection system between the CNS and gastrointestinal tract based on the vagus nerve, enteric nervous system, neuroendocrine system, and circulatory system, thereby affecting the gut microbiota homeostasis and brain function, including behavior. The Gut–Lung–Brain (pink) axis is an intricate network, linking the gut, lung, and brain, that consists of various components, such as vagus nerve, hypothalamus–pituitary–adrenal (HPA) axis, immune system, metabolites, and bacterial microbiota. For instance, the vagus nerve, communicating with the gastrointestinal tract and respiratory apparatus, influences the motility, immunity, permeability of gut mucosa and bronchial smooth muscle contraction, and oxygen consumption. The oral–brain–gut axis (orange) is a complex interconnection among the oral cavity, brain, and gut, and it is mostly observed by studying the role of oral microbiota in periodontitis and neurodegenerative diseases. Microbiota resident in the gut–lung epithelial mucosa are among the targets of these molecules, and, in turn, they respond by producing different mediators (such as fatty acids, gut peptides, BEVs) that impact directly and indirectly the brain functions. The brain–gut–other organ axis (blue) is mainly based on the vagus nerve’s innervation of other organs. In particular, the vagus nerve supplies parasympathetic fibers to all organs, except the adrenal glands, transmitting and receiving information feedback. (Created with BioRender.com).

**Figure 3 ijms-25-08722-f003:**
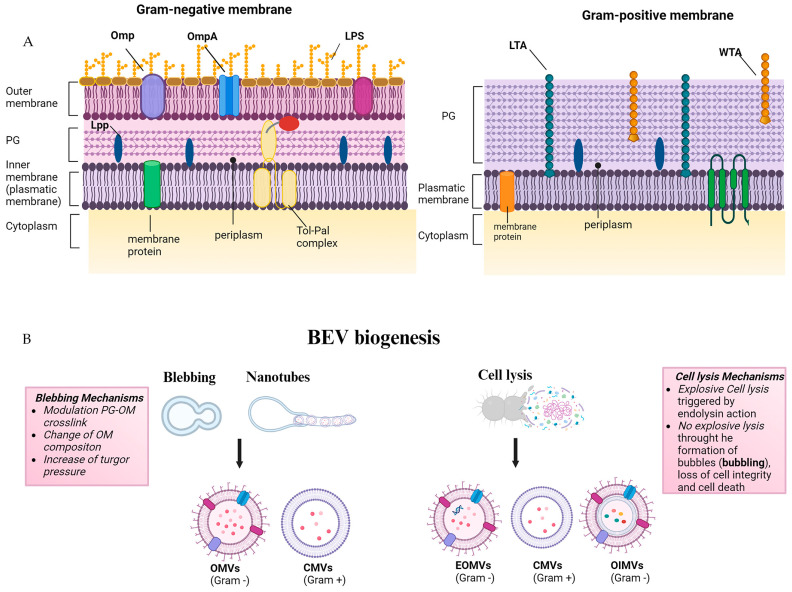
Structure of the Gram-negative and Gram-positive cell envelope and biogenesis mechanisms of BEVs. (**A**) The architecture of Gram-negative cell envelope consists of two membranes: outer membrane (OM) and inner membrane (IM). The OM consists of an exterior leaflet of lipopolysaccharides (LPS) and an internal leaflet of phospholipids while IM is composed of a classic phospholipid bilayer. Between IM and OM, there is the periplasmic space, a thin layer of peptidoglycans (PG) in which Braun’s lipoproteins (Lpp) are immersed and covalently link PG to the two layers providing structural integrity to OM. The porin outer-membrane proteins (Omp) and Tol–Pal (peptidoglycan-associated lipoprotein) complex are embedded in OM and interact with OM via PG. The structure of the Gram-positive cell envelope consists of a thick layer of PG in which are present molecules of lipoteichoic acids (LTA) covalently linked to lipids of the underlying cytoplasmic membrane and wall lipoteichoic acids (WTA), conferring a negative charge to Gram-positive bacteria. The plasmatic membrane (PM) is a classic lipid bilayer in which are immerse membrane channels and functional transmembrane proteins (in orange and green colors). The periplasmic space is located between PG and PM. (**B**) BEV biogenesis occurs through three mechanisms: blebbing, explosive cell lysis, and nanotube formation. Gram-negative bacteria produce mainly OMVs through blebbing and EOMVs through explosive cell lysis in which OM dissociates from the PG, forming OM vesicles; OIMVs are produced by explosive cell lysis and contain both inner and outer membranes. Gram-positive bacteria produce cytoplasmic membranes (CMVs) lacking an OM through an explosive cell lysis mechanism or a non-explosive lysis (bubbling) consisting of the cell integrity loss and cell death. The formation of BEVs from nanotube, filamentous structures occurs in both Gram-positive bacteria, through a process of extrusion of the plasma membrane, and Gram-negative bacteria, from the extrusion of the OM. (Created with BioRender.com).

**Figure 4 ijms-25-08722-f004:**
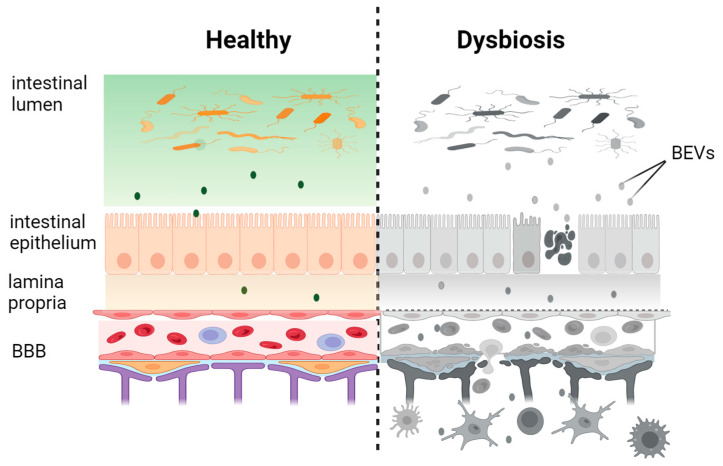
Schematic interaction between BEVs and cellular barriers in health and disease (created with BioRender.com).

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
