# Peer review of "Microbiota-Derived Extracellular Vesicle as Emerging Actors in Host Interactions"

_ijms, 2024, doi:10.3390/ijms25168722_

Round 1

Reviewer 1 Report

Comments and Suggestions for Authors

It was well-written and provided further information regarding microbiota-derived extracellular vesciles.

In the introduction section, the author can provide enough background on microbiota and BEVs in host interactions.

In the end of discussion section the author can expand on BEV research's ramifications and future uses. Based on your results, please describe the BEV-influenced pathways or processes and recommend future study or uses. 

If possible, the author can add one more figure for mechanistic explanation.

Author Response

Reviewers 1

Comments and Suggestions for Authors

It was well-written and provided further information regarding microbiota-derived extracellular vesciles. In the introduction section, the author can provide enough background on microbiota and BEVs in host interactions. In the end of discussion section the author can expand on BEV research's ramifications and future uses.

  • Based on your results, please describe the BEV-influenced pathways or processes and recommend future study or uses. If possible, the author can add one more figure for mechanistic explanation.

Response to Reviewer

Our study focuses on human extracellular vesicles as biomarker source in multiple sclerosis (“Extracellular vesicles as contributors in the pathogenesis of multiple sclerosis”, Mult Scler Relat Disord. 2023 Mar:71:104554. doi: 10.1016/j.msard.2023.104554; Proteomic profile of extracellular vesicles from plasma and CSF of multiple sclerosis patients reveals disease activity-associated EAAT2”has just accepted for publication by Journal of Neuroinflammation). Recently, our research interest has addressed to bacterial extracellular vesicles as potential players in multiple sclerosis pathogenesis. According your suggestion, we described deeply the mechanisms by which BEVs interact with epithelium of different part of the body, such as gut, oral cavity, lung and skin, and act as immunomodulators in presence of viral infection. Moreover, we added two more figures: figure 3, in which structure of the Gram-negative and Gram-positive cell envelope and biogenesis mechanisms of BEVs are depicted; figure 4, where schematic interaction between BEVs and cellular barriers in health and disease are shown.

Reviewer 2 Report

Comments and Suggestions for Authors

The current narrative review entitled "Microbiota-derived extracellular vesicles as emerging actors in host interactions" examines the role of microbiota-derived EVs from the gut and oral cavity in the interactions between the epithelium, immune system and CNS. I have some comments: - please discuss the methods used to quantify or analyze these EVs, including costs and whether they can possess any practical applicability - what is the clinical application of your findings? can these EVs serve a clinical purpose or are they used only for research? Unfortunately I believe that a similarity index of 58% according to iThenticate is too high, the similarity with your text and the first and second identified sources are 15% and 8%, respectively. The authors should try to rephrase the paper and rewrite some of its parts in their own words, adding a critical note to the paper and their original touch, otherwise it is just a compilation of text.       

Author Response

Reviewer 2

The current narrative review entitled "Microbiota-derived extracellular vesicles as emerging actors in host interactions" examines the role of microbiota-derived EVs from the gut and oral cavity in the interactions between the epithelium, immune system and CNS. I have some comments: - please discuss the methods used to quantify or analyze these EVs, including costs and whether they can possess any practical applicability - what is the clinical application of your findings? can these EVs serve a clinical purpose or are they used only for research? Unfortunately I believe that a similarity index of 58% according to iThenticate is too high, the similarity with your text and the first and second identified sources are 15% and 8%, respectively. The authors should try to rephrase the paper and rewrite some of its parts in their own words, adding a critical note to the paper and their original touch, otherwise it is just a compilation of text.      

  • Please discuss the methods used to quantify or analyze these EVs, including costs and whether they can possess any practical applicability - what is the clinical application of your findings?

Response to Reviewer

To date we have not published any scientific papers on BEVs, apart this review, although we currently study human extracellular vesicles (our article “Proteomic profile of extracellular vesicles from plasma and CSF of multiple sclerosis patients reveals disease activity-associated EAAT2”has just accepted for publication by Journal of Neuroinflammation). Therefore, we are not able to provide information you requested, but we can give you bibliographic reference about that : Tulkens, J.; De Wever, O.; Hendrix, A. Analyzing bacterial extracellular vesicles in human body fluids by orthogonal bi-ophysical separation and biochemical characterization. Nat Protoc. 2020, 15, 40–67. doi: 10.1038/s41596-019-0236-5. This reference was also reported in our review at number 101.

  • Unfortunately I believe that a similarity index of 58% according to iThenticate is too high, the similarity with your text and the first and second identified sources are 15% and 8%, respectively. The authors should try to rephrase the paper and rewrite some of its parts in their own words, adding a critical note to the paper and their original touch, otherwise it is just a compilation of text.

Response to Reviewer

We changed some sentences in our text, but due to the lack of the iThenticate software we did not have the possibility to check it for similarity index calculation.

Generally, the review articles could have major similarity index score than research articles due to the fact that the review articles report previous published results from the scientific literature meanwhile the original research articles report new findings. Anyway, for every phrase, comment and concept, we had always attributed the authorship, indicating the bibliographic reference.

Reviewer 3 Report

Comments and Suggestions for Authors

This report by Paola Margutti et al. summarizes the effects of bacterial extracellular vesicles on the brain. In recent reports on the analysis of EVs, there are many research reports that bridge previously unknown fields, and this report summarizing them is very significant. While many of the EVs are cancer research, there is also a lot of data suggesting a relationship between the intestine and the brain, which is essential for reviewing recent reports, and this report summarizing the latest information is thought to be useful not only in specialized fields but also in many other fields.

There is no problem with the research argument or text structure. However, I think that figures should be made easier for readers to understand.

For example, in Figure 1, it is difficult to understand the relationship between the Microbiota on the left and the nervous system on the right. It seems that the colors of the oral cavity and digestive tract are different from the colors of other organs, but the colors of the other organs and the Microbiota are similar colors. I think it would be better to make it easier to understand.

Similarly, the figure on page 18 is also called Figure 1. Shouldn't this be Figure 2? Regarding color usage, generally, when the condition changes from healthy to dysbiosis, dark colors are often used. The authors do the same for the intestines. On the other hand, bright colors are used for bacteria. This does not convey the image to the reader. The same goes for the colors of vascular endothelial cells. Please consider these points and reconsider.

Author Response

Reviewer 3

This report by Paola Margutti et al. summarizes the effects of bacterial extracellular vesicles on the brain. In recent reports on the analysis of EVs, there are many research reports that bridge previously unknown fields, and this report summarizing them is very significant. While many of the EVs are cancer research, there is also a lot of data suggesting a relationship between the intestine and the brain, which is essential for reviewing recent reports, and this report summarizing the latest information is thought to be useful not only in specialized fields but also in many other fields.

There is no problem with the research argument or text structure.

  • However, I think that figures should be made easier for readers to understand. For example, in Figure 1, it is difficult to understand the relationship between the Microbiota on the left and the nervous system on the right. It seems that the colors of the oral cavity and digestive tract are different from the colors of other organs, but the colors of the other organs and the Microbiota are similar colors. I think it would be better to make it easier to understand. Similarly, the figure on page 18 is also called Figure 1. Shouldn't this be Figure 2? Regarding color usage, generally, when the condition changes from healthy to dysbiosis, dark colors are often used. The authors do the same for the intestines. On the other hand, bright colors are used for bacteria. This does not convey the image to the reader. The same goes for the colors of vascular endothelial cells. Please consider these points and reconsider.

Response to reviewer

According your suggestion, we made changes to the figures 1 and 2, trying to explain more comprehensively in their legends. Moreover, we added two more figures: figure 3, in which structure of the Gram-negative and Gram-positive cell envelope and biogenesis mechanisms of BEVs are depicted; figure 4, where schematic interaction between BEVs and cellular barriers in health and disease are shown.

Reviewer 4 Report

Comments and Suggestions for Authors

This is an extensive and comprehensive review that examines a recent "player" in the microbiome story, that of "bacterial extracellular vesicles (BEV). BEV's are very small out pouching of bacterial membranes that can penetrate classical cell membrane barriers, including the blood-brain barrier (BBB), carry their cargoes into somatic cells and theoretically influence pathological processes in the brain (in addition to other bodily organs). The authors extensively review current knowledge about BEV's , how they form, which bacteria are known to form them, and what effect(s) they have on target cells they enter. They include at the end of the paper a section on BEV's and the brain, which I found particularly informative. 

I feel that the paper can be published as is, but I would like to suggest to the authors that they carry out a minor modification to their otherwise very comprehensive and informative text.

I suggest using more frequent paragraph indentations to "break up" the knowledge communicated to smaller subjects areas. This approach is more a matter of style than substance. In addition, I feel that the paper would also benefit from a small table that lists the bacteria known to make BEV's.

I feel that the Figures included are not only well drawn and easy to interpret, but contribute to the complex topics presented.

Comments on the Quality of English Language

no changes needed

Author Response

Reviewer 4

This is an extensive and comprehensive review that examines a recent "player" in the microbiome story, that of "bacterial extracellular vesicles (BEV). BEV's are very small out pouching of bacterial membranes that can penetrate classical cell membrane barriers, including the blood-brain barrier (BBB), carry their cargoes into somatic cells and theoretically influence pathological processes in the brain (in addition to other bodily organs). The authors extensively review current knowledge about BEV's , how they form, which bacteria are known to form them, and what effect(s) they have on target cells they enter. They include at the end of the paper a section on BEV's and the brain, which I found particularly informative.

I feel that the paper can be published as is, but I would like to suggest to the authors that they carry out a minor modification to their otherwise very comprehensive and informative text.

1)            I suggest using more frequent paragraph indentations to "break up" the knowledge communicated to smaller subjects areas. This approach is more a matter of style than substance. In addition, I feel that the paper would also benefit from a small table that lists the bacteria known to make BEV's.

Response to reviewer

According your suggestion, we made breaks in the paragraphs and we inserted a table where bacterial species producing BEVs, reported in the manuscript, are listed with relative references.

Round 2

Reviewer 2 Report

Comments and Suggestions for Authors

The authors have answered to my comments